# Evolutionary origins and functional diversification of Auxin Response Factors

Jorge Hernández-García [1], Vanessa Polet Carrillo-Carrasco[1,4], Juriaan Rienstra [1,4], Keita Tanaka[1,3], Martijn de Roij [1], Melissa Dipp-Álvarez [1], Alejandra Freire-Ríos[1], Isidro Crespo [2], Roeland Boer [2], Willy A. M. van den Berg[1], Simon Lindhoud[1] & Dolf Weijers [1] ✉

The Auxin Response Factors (ARFs) family of transcription factors are the central mediators of auxin-triggered transcriptional regulation. Functionally different classes of extant ARFs operate as antagonistic auxin-dependent and -independent regulators. While part of the evolutionary trajectory to the present auxin response functions has been reconstructed, it is unclear how ARFs emerged, and how early diversification led to functionally different proteins. Here, we use in silico and in vivo analyses to revisit the molecular events that led to the origin and subsequent evolution of the ARFs. We reveal the shared origin of ARFs from preexisting domains, uncovering a protein fold homologous to the ARF DNA-binding fold in a conserved eukaryotic chromatin regulator. Building on this, we reconstruct the complete evolutionary history of ARFs, including the divergence events leading to the appearance of the ARF classes and defining the main molecular targets for their functional diversification. We derive a complete evolutionary trajectory that led to the emergence of the nuclear auxin signalling pathway.

Auxin regulates plant development and acts as a communication signal between cells, tissues and organisms[1]. Its activity in controlling growth and development prominently involves transcriptional regulation through the nuclear auxin pathway (NAP). This signalling pathway is conserved in land plants and functions through three dedicated elements: The F-box family of TIR1/AFB receptors, which interact with the second element upon auxin binding, the Aux/IAA co-repressors, and destabilize them; thus releasing the final effectors, the Auxin Response Factors (ARF)[2]. Some members of the ARF family of DNA-binding transcription factors (TF) play an essential role in auxin responses, as they contribute target selectivity through DNA-binding, and transcriptional regulatory activity, while the receptor and co-repressor act as the permissive, auxin-dependent regulatory module for these effectors[3].

ARF domain architecture includes an N-terminal DNA-binding domain (DBD), a C-terminal oligomerization PB1 domain, and an intrinsically disordered middle region (MR) between them[4]. DBDs harbor three independently folded subdomains, the Viridiplantae-specific B3 DNA-binding domain, the dimerization domain (DD), and the ancillary domain (AD)[5]. While the B3 domain, that mediates direct DNA-binding, can be found in other subfamilies of plant TF such as RAVs or ABI3-related proteins[6], the DD and AD folds are considered ARF-specific. These two subdomains are thought to act as dimerization and structural scaffolds for cooperative DNA-binding, respectively. PB1 domains are present in all eukaryotes in different protein families, and are commonly involved in head-to-tail oligomerization, an essential feature in auxin response by driving ARF-Aux/IAA interaction[7].

[1]Laboratory of Biochemistry, Wageningen University, Stippeneng 4, 6708WE Wageningen, the Netherlands. [2]Experiments Division, ALBA Synchrotron Light Source, Carrer de la Llum 2–26, 08290 Cerdanyola del Vallè's, Catalunya, Spain. [3]Present address: CAS Center for Excellence in Molecular Plant Sciences, 300 Feng Ling Road, Shanghai 200032, PR China. [4]These authors contributed equally: Vanessa Polet Carrillo-Carrasco, Juriaan Rienstra. ✉e-mail: dolf.weijers@wur.nl

While auxin is present in algae and non-plant organisms[8], the NAP can only be found in land plants, suggesting it appeared during plant terrestrialization[9–11]. The distant TIR1/AFB homologs in algae lack critical auxin-binding residues, coinciding with the absence of TIR1-interacting motifs in the Aux/IAA homologs. This contrasts with the widespread presence of ARFs in streptophyte algae, the closest relatives to land plants. During their evolution, ARFs have diverged into three functionally different classes: A, B and C[12]. A- and B-ARFs share DNA-binding specificity, while C-class specificity is unknown, but different from A- and B-classes'[13]. The A-ARFs are the only class regulated by auxin through the NAP and acting as transcriptional activators. In opposition, B- and C-ARFs are considered transcriptional repressors. Analysis in the liverwort *Marchantia polymorpha* showed that the minimal auxin response requires the antagonistic interplay between an auxin-regulated A-ARF activator, and a B-ARF repressor competing for the same targets[14]. In contrast, the Marchantia C-ARF appears not to be associated with the direct regulation of auxin responses[11,15].

While phylogenomic and genetic analysis have given deep insights into the origin and minimal architecture of the nuclear auxin response system, important questions remain unanswered. First, it is unclear what innovations led to the appearance of the multi-domain ARF transcription factors. Second, the path from ARF appearance to the functional diversification in A, B and C classes is unclear, as well as if extant ARF proteins have retained ancestral functions or capacities. Here, we re-evaluate ARF evolution using the latest available genomic data. Our approach identifies the ancestral origin of the ARF DBD fold in the Tudor-like domains of a family of eukaryotic chromatin reader proteins. We confirm the close relationship of the remaining ARF domains with those of plant-specific RAV proteins, and find that at least two different genetic sources combined to form extant ARF architecture. Through genetic and biochemical approaches, we further infer the most likely evolutionary path that ARFs followed from their origin until they became an essential part of the auxin response systems. Thus, we offer a complete perspective on the birth and rise of the auxin response effectors.

## Results

### The origin of the DNA binding domain fold in Auxin Response Factors

Previous attempts to find distant ARF homologs resulted in AD-like sequences in red algae and chlorophytes, but these occurrences were scattered, and it was not possible to pinpoint the source of the ARF domains[11]. Initial exploration of homologies suggested resemblance of part of the ARF DBD to Bromodomain-WP40 domains[16], but the nature of this domain was not clear. We revisited the existence of ARF-like proteins focusing on the ARF AD amino acid sequence by HMMER searches in the UniProt Reference Proteomes database. Using *Marchantia polymorpha* ARF1 AD or *Chlorokybus melkonianii* ARF AD, we retrieved a list of proteins in which 90% represented ARF proteins. The remaining proteins were non-ARF proteins belonging to diverse eukaryotic lineages (Supplementary Data 1). These hits included the red algae *Chondrus crispus* homologous AD sequence previously identified. Using this sequence as phmmer bait resulted in 2472 sequences, of which half (54%) were strictly of streptophyte origin and reflected the stereotypical ARF architecture with a B3 (PF02362)-containing DBD, a middle region (MR), and an Aux/IAA-type Phox and Bem1 (PB1) domain (PF02309) (Fig. 1a, Supplementary Data 1). The remaining sequences (45%) were distributed across different eukaryotic lineages, including Archaeplastida and Opisthokonta, with architectures defined by an N-terminal WD40 propeller (PF00400) composed of seven or eight WD40 repeats, and one or two C-terminal bromodomains (BRD, PF00439). These sequences were orthologous to the PHIP/BRWD family of chromatin-associated proteins known to bind methylated histones in metazoans[17]. We

repeated the search using the human PHIP homologous region with similar results (Supplementary Data 1). Both ARF and PHIP architectures shared the putative presence of a predicted Auxin response factor-type domain annotation (Auxin_resp, PF06507, Fig. 1a), partially coinciding with the ARF AD domain (ARF[AD]). However, this feature falls below the cut-off range for identification in PHIPs. A manual search in PHIP Auxin_resp flanking regions exposed an extended region of shared homology with the entire ARF[AD] sequence plus the upstream DD. In contrast to ARFs, the region that would be occupied by the B3 domain was filled by short sequences of low conservation (Supplementary Fig. 1). Alignment between these homology regions reflected remarkable conservation in primary sequence, supporting shared ancestry (Fig. 1b). Phylogenetic analysis indicated that the ARF[DD-AD] sequence likely originated from a PHIP-related sequence within the Archaeplastida lineage (Fig. 1c), given its closeness to glaucophytes and rhodophytes (red algae). As PHIP/BRWDs are widespread and deeply conserved among eukaryotes, these proteins likely evolved in a last eukaryote common ancestor, indicating that the PHIP origin vastly predates that of the streptophyte-specific ARF proteins.

The DD-AD homology region in PHIPs corresponds to the crypto-Tudor (cTudor) domain of human PHIP (PHIP[cTudor]), a functional histone-reading double Tudor-like domain[17]. Initial ARF[DBD] structure observations noted the resemblance between ARF[AD] and Tudor-like folds[5]. A re-analysis of ARF[DBD] structure indicated that the DD fold (R40-P99 and S238-276 in Arabidopsis ARF1, PDB: 4ldx) is similar to that of Royal Family domain folds as Chromodomains or other Tudor-like domains, and matches that of the first Tudor domain of PHIP[cTudor] but with an inserted B3. The second PHIP[cTudor] fold perfectly overlaps with the ARF[AD], showing an overall Root Mean Square Deviation (RMSD) of 1.10 Å between ARF and PHIP DD-AD regions (Fig. 1d, Supplementary Fig. 2a). This structural similarity further supports the evolutionary relationship between PHIP[cTudor] and the DD-AD fold in ARF[DBD]s. However, there are differences between both structures, such as the lack of a B3 in PHIP[cTudor]. A key ARF feature is the DD α-helix 6 (in MpARF1) that facilitates dimerization in two-fold rotational (C2) symmetry[5,14]. This α-helix is predicted to be absent in the PHIP[cTudor] structure, which suggests it was specifically acquired in the ARF lineage.

Tudor-like and other Royal Family domains contain a hydrophobic cage (HC) with specific hydrophobic residues involved in binding methylated histones. Our analyses indicated that ARF[AD] fold conserves such residues in the same position as those found to be functional for the second Tudor domain in PHIP[cTudor], while both ARF[DD] and the first PHIP Tudor-like repeat have unclear HCs (Fig. 1e, Supplementary Fig. 2b), indicating this upstream fold may act as an extended structural fold involved in stabilizing the main histone binder domain. However, ARF[AD] HC is blocked by a loop through conserved positively charged residues (Fig. 1e, j). Similarly, the partial ARF[DD] HC could be partly blocked by α-helixes 2 and 7, but this is also possible for PHIPs, with the equivalent of α-helix 2 (Supplementary Fig. 2c). Altogether, our analyses suggest that the cTudor and DBD structures are closely related and formed by a histone reader Tudor-type double-domain fold structure. However, it remains unclear whether ARF[DD] and/or ARF[AD] hold additional functions as histone readers.

As suggested by the structure, perturbing ARF dimerization through the DD renders proteins non-functional[5,18]. However, ARF[AD] function has not been assessed, prompting us to analyse the physiological relevance of this subdomain. We assessed this in two different and distant experimental models, *Arabidopsis thaliana* ARF5/MONOPTEROS and *Marchantia polymorpha* ARF1. Knock-out mutants of these genes have conspicuous phenotypes compared to wild-type plants that can be complemented with the wild-type genes (Fig. 1f-i). Deleting the AD in a way that the DD-B3 structure should

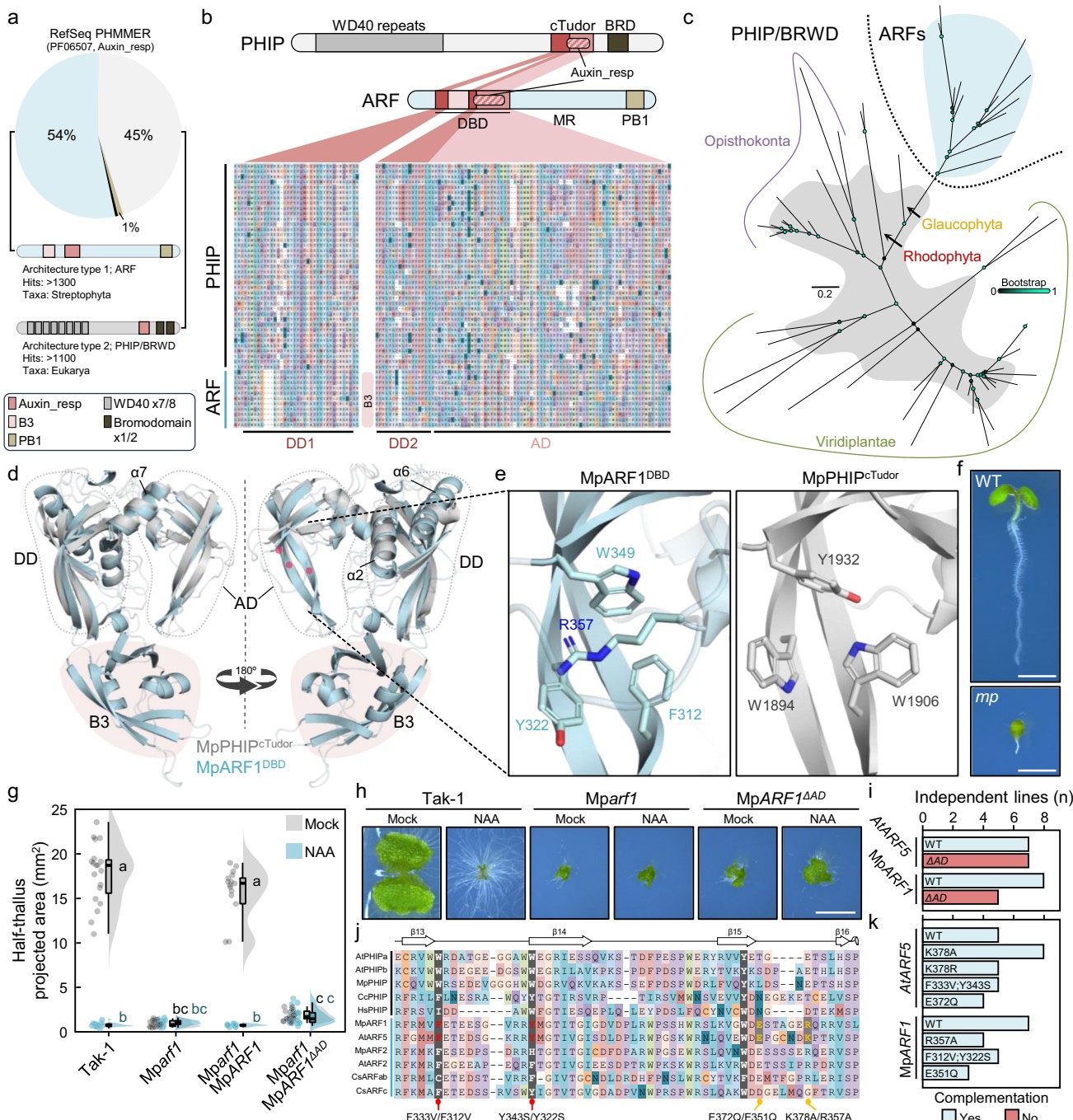

**Fig. 1 | The ARF DNA-binding scaffold evolved from an ancestral Tudor-like domain. a** PHMMER hits using DD-AD region on the UniProt Reference Proteomes database. Summary of architectures found merged into either ARF or PHIP are shown. **b** PHIP and ARF architecture comparison and multiple protein sequence alignment of the double Tudor domain regions based on *Marchantia polymorpha* protein primary sequences of MpPHIP and MpARF1. **c** Unrooted maximum likelihood gene tree of the Tudor-like regions of ARFs and PHIPs. Bootstrap values are indicated as color-coded bubbles in branch nodes. Scale bar indicates substitutions per residue. **d** Structural alignment of the DD-AD of PHIP and ARF, represented by *Marchantia polymorpha* PHIP and ARF1 predicted structure. Blue, MpARF1; Grey, MpPHIP. Key alpha helixes are pointed and DD, B3 and AD domains marked. Red circles represent the position of the hydrophobic cage residues in AD. **e** Magnification of the hydrophobic cage structure characteristic of Tudor domains in MpPHIP and MpARF1 with key amino acids highlighted. **f** *Arabidopsis thaliana* wild-type (WT) and *mp/arf5* mutant seedlings used to score complementation in **i**.

Scale bar, 2 mm. **g** Raincloud plot of half-thallus area measurements of 10-day-old Mp*arf1* mutant complemented lines with AD-deleted versions grown in mock (DMSO) or auxin (3 μM NAA). *n* = 19,19,17,20,17,20 (left to right, mock/treatment). Boxplots in Raincloud indicate the following parameters: centrum, median; upper bound, first quartile; lower bound, third quartile; whiskers maximum and minimum refer to highest and lowest values, respectively, within 1.5*inter-quartile range (IQR). Statistical groups are determined by Tukey's Post-Hoc test (*p* < 0.05) following one-way ANOVA. **h** Representative pictures of *M. polymorpha* plants in (**g**), Scale bar, 5 mm. **i** Summary of AtARF5 (MONOPTEROS, MP) and MpARF1 complementation of each corresponding mutant per transgenic line with AD deleted versions. **j** Multiple protein sequence alignment of the AD region on selected PHIP and ARF proteins showing hydrophobic cage residues, highlighting mutations for analyses in **k** referring to AtARF5 and MpARF1 residue positions. **k** AtARF5 and MpARF1 complementation of each corresponding mutant per transgenic line with hydrophobic cage point mutants. Source data are provided as a Source Data file.

not be affected impaired mutant complementation in several independent lines in both species, indicating that the AD is essential for ARF function (Fig. 1g-i, Supplementary Fig. 3). Given its evolutionary link with a histone reader, and the presence of a conserved HC, it is plausible that it acts as a histone binder. To test the role of HC residues and – by extension – the potential for histone binding, we introduced individual mutations known to disrupt Tudor-domain HC pockets in AtARF5 and MpARF1 (Fig. 1j). None of the mutations reduced the ability to complement the mutant phenotypes, indicating that the putative HC is not necessary for the function of ARF proteins, and suggesting that histone binding is not a physiologically relevant function of ARF[AD] (Fig. 1k, Supplementary Fig. 4). Instead, our results indicate that ADs have an unknown integral function within the DBD different from methyl-histone binding. As initially suggested based on exploratory homology analysis[16], we propose that the ARF[DBD] domain evolved from a functional histone reading cTudor-domain of a PHIP homolog that integrated a B3 DNA-binding domain within its sequence in a streptophyte ancestor.

## Multiple RAV and ARF domains share recent ancestry

ARF proteins share domains with the Related to ABI3 and VP1 (RAV) subfamily of B3 TFs. The domain architecture of both subfamilies has three unambiguously shared features: the B3 DNA binding domain, the PB1 domain, and the B3 repression domain (BRD, also known as LFG)[10,11,15,19]; (Fig. 2a). However, whether both subfamilies derive from a recent common ancestral gene is unknown. B3 domains evolved from a restriction enzyme DNA-recognition fold[20], and are found in other plant TFs of the ABI3, HSL, and REM subfamilies. B3 domains (hereafter referred as TF[B3]s) from RAVs and ARFs have been suggested to be related[10,11], but the extent of this relation is unclear. While B3 domains share a common structure of a pseudo-barrel composed of seven β-strands and two α-helixes, primary sequence conservation is partial, leading to uncertain evolutionary relationships within the family[6,11,21]. To better understand the relationships among B3 members, we surveyed a range of genomic resources, including independent genome projects, transcriptomes and updated annotations. We confirmed that only Viridiplantae species contain bona fide B3 domains

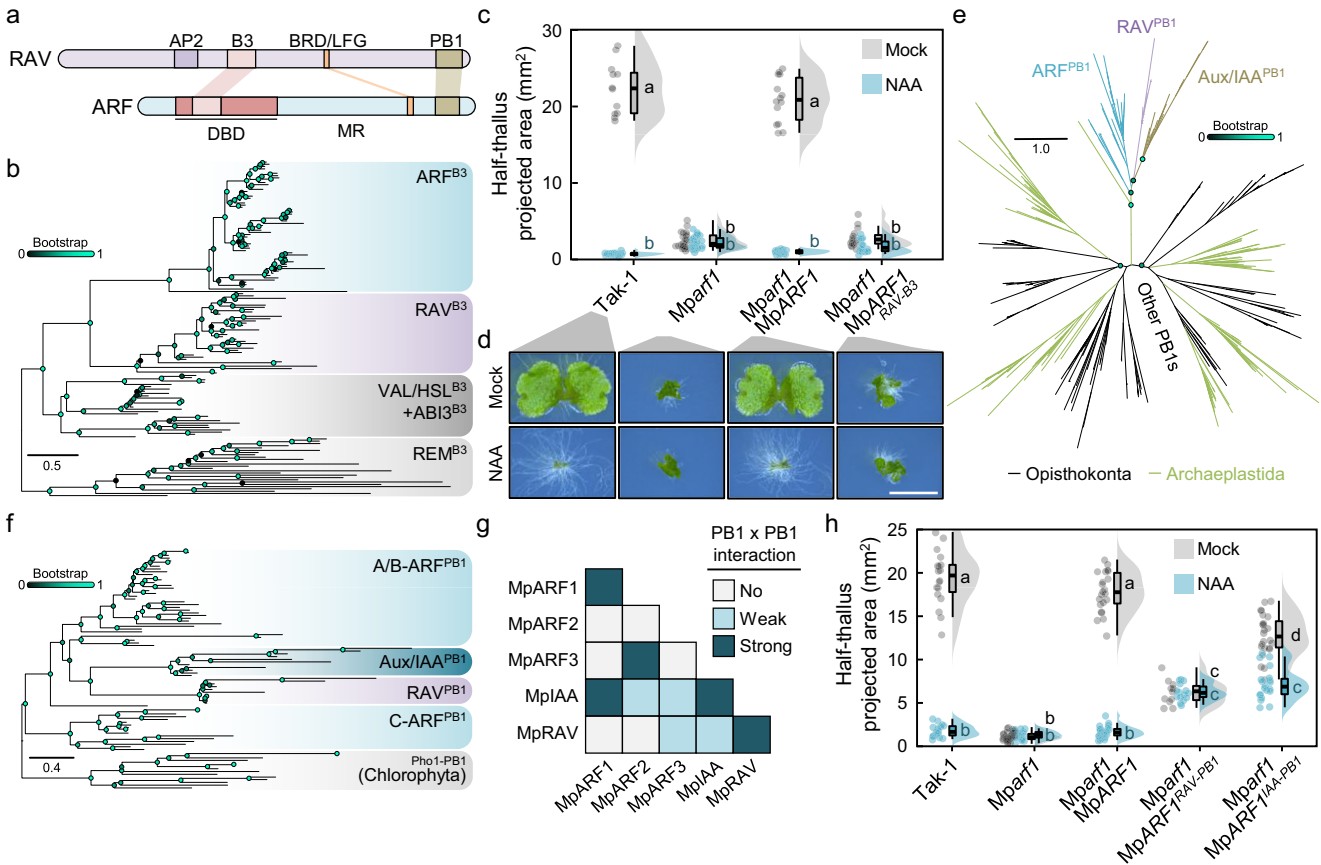

**Fig. 2 | RAVs and ARFs are ancestral streptophyte-specific ortholog genes. a** Schematic representation of RAV and ARF proteins primary sequence based on *Marchantia polymorpha* protein primary sequences of RAV and ARF1. BRD stands for B3 repression domain. **b** Maximum likelihood square phylogenetic tree of B3 domains in Viridiplantae rooted using REM as outgroup. Bootstrap values are indicated as color-coded bubbles in branch nodes. Scale bar represents distance in substitutions per residue. **c** Raincloud plot of half-thallus area measurements of 10-day-old Mp*arf1* mutant complemented a line with MpARF1 B3 domain swapped for MpRAV B3 grown in mock (DMSO) or auxin (3 μM NAA). *n* = 16,15,24,28,14,19,16,15 (left to right, mock/treatment). Statistical groups are determined by Tukey's Post-Hoc test (*p* < 0.05) following one-way ANOVA. **d** Representative pictures of plants in **c**), Scale bar, 5 mm. **e** Unrooted maximum likelihood gene tree of the eukaryotic PB1s re-analysed from Mutte & Weijers 2020. Bootstrap values are indicated in key nodes as color-coded bubbles in branch nodes. Scale bar represents distance in substitutions per residue. **f** Maximum likelihood square phylogenetic tree of ARF/ RAV-related PB1 domains rooted using Chlorophyta Pho1-like PB1 sequences as outgroup. Bootstrap values are indicated as color-coded bubbles in branch nodes. Scale bar represents distance in substitutions per residue. **g** Summary of *Marchantia polymorpha* PB1 pairwise interaction assays in yeast-two-hybrid combining drop and galactosidase activity assays (see Supplementary Fig. 6, and Supplementary Data 2). **h** Raincloud plot of half-thallus area measurements of 10-day-old Mp*arf1* mutant complemented lines with MpARF1 PB1 domain swapped for MpRAV and MpIAA PB1s grown in mock (DMSO) or auxin (3 μM NAA).
*n* = 18,17,27,24,23,28,11,15,24,20 (left to right, mock/treatment). Boxplots in Raincloud indicate the following parameters: centrum, median; upper bound, first quartile; lower bound, third quartile; whiskers maximum and minimum refer to highest and lowest values, respectively, within 1.5*inter-quartile range (IQR). Statistical groups are determined by Tukey's Post-Hoc test (*p* < 0.05) following one-way ANOVA. Source data are provided as a Source Data file.

(Supplementary Fig. 5a). ABI3/HSL-like proteins were assumed to be reminiscent of an ancestral B3 protein based on the presence of well-conserved orthologs in chlorophytes[20]. We also found a well-supported REM clade containing chlorophyte sequences, suggesting that both REM and ABI3/HSL-related B3 subfamilies were present in a Viridiplantae common ancestor. In turn, we found RAV[B3] and ARF[B3] only in streptophytes, suggesting that these domains appeared with RAV and ARF genes in a streptophyte common ancestor. The divergence of REM[B3]s primary sequences prompted us to use this group as outgroup to resolve the relationships between ABI3/HSL, RAV and ARF (Fig. 2b, Supplementary Fig. 5b). The rooted tree indicated that RAV[B3]s and ARF[B3] form a unique clade within the B3 family, suggesting that RAV[B3] and ARF[B3] diverged from an ancestral ARF/RAV[B3]. Extant RAV[B3] and ARF[B3] bind different DNA cis elements[10], yet B3 structures resemble each other (Supplementary Fig. 5c–e). Binding site specificity is linked to B3 sequence, but the context of each B3 domain may contribute to it, possibly underlying specificity divergence after RAV[B3] and ARF[B3] split. It is thus plausible that RAV[B3] can still partly function as an ARF[B3] if embedded within the DD-AD fold of ARF[DBD]. To test this, we performed Marchantia polymorpha Mparf1 and Mparf3 mutant complementation assays using native ARF genes in which the B3 sequences had been seamlessly swapped for the MpRAV[B3] sequence. Neither MpARF1[RAV-B3] nor MpARF3[RAV-B3] were able to rescue their respective mutant phenotypes (Fig. 2c, d, Supplementary Fig. 6a, b). This indicates that the B3 endows DNA-binding specificity in ARFs, and this specificity likely diverged soon after RAV[B3] and ARF[B3] split.

Bryophytes and streptophyte algae, but not tracheophytes, contain PB1-bearing RAV orthologs[10,15], suggesting that RAV[PB1] was lost in a vascular plant common ancestor. RAV[PB1]s are closely related to ARF and Aux/IAA PB1s and likely originated in a streptophyte common ancestor (Fig. 2e)[7]. To establish a scenario of ARF-Aux/IAA-RAV PB1 evolution, we expanded previous analyses with additional sequences including the Chlorophyta Phox1-like[PB1]s as the closest relative PB1 clade (Fig. 2f). Our phylogenetic tree confirmed the monophyly of ARF-Aux/IAA-RAV PB1s, as well as showed a maximum-statistically supported clade of Aux/IAA[PB1] and RAV[PB1]s robustly separated from the ARF[PB1] sequences. Given the presence of RAVs in all streptophyte algae, but Aux/IAA likely appearing only later in evolution, this analysis suggests that Aux/IAA[PB1]s could have evolved from RAV[PB1]s, however both clades show long divergences, potentially affecting tree topology. RAV[PB1]s are type I/II PB1 domains containing identifiable positively and negatively charged interfaces that can oligomerize as ARF[PB1]s[10]. Given the close relationship between these PB1s, it is plausible for RAV[PB1]s to function as ARF or Aux/IAA PB1s. To test this, we assayed all pairwise combinations between M. polymorpha ARF, Aux/IAA and RAV[PB1]s in a yeast two-hybrid assay (Fig. 2g, Supplementary Fig. 6d, Supplementary Data 2). We observed strong homo- and heterotypic interactions between MpARF1[PB1] and MpIAA[PB1], while MpARF2[PB1] and MpARF3[PB1] only showed a strong heterotypic interaction between them. In turn, MpRAV[PB1] strongly self-interacted, indicating potential oligomerization. A weak interaction was observed in a single direction between MpRAV[PB1] and both MpARF3's and MpIAA's, similar to B- and C-class ARF[PB1] behaviour. To explore if MpRAV[PB1] homotypic interaction reflects a biologically relevant property, we performed PB1 domain swap of MpARF1[PB1] and MpRAV[PB1]s (MpARF1[RAV-PB1]) in a Mparf1 complementation assay. MpARF1 requires its PB1 to oligomerize for its intrinsic activity, but also to be auxin-regulated through the interaction with MpIAA[PB1]. All the lines expressing MpARF1[RAV-PB1] chimeras showed mild but noticeable complementation of the Mparf1 mutant phenotype (Fig. 2h, Supplementary Fig. 6e, f), indicating that MpRAV[PB1] is able to oligomerize in a cellular context. These plants did however not respond to auxin in terms of growth inhibition, in line with the lack of strong interaction between MpRAV[PB1] and MpIAA[PB1]. In contrast, a MpIAA[PB1] swap into MpARF1 (MpARF1[IAA-PB1]) partly complemented not only the growth phenotype but also auxin responsiveness, confirming

its ability to homo- and heterotypic oligomerization, as expected from its interaction profile. This indicates that RAV[PB1]s share ancestry and partial functionality with ARF[PB1]s, but interaction abilities have evolved and refined during evolution to favour specific PB1-to-PB1 interactions.

Based on phylogenetic re-evaluation and genetic and interaction assays, we therefore propose that RAV and ARF evolved from a common RAV/ARF ancestral gene.

## PHIP and RAV are not involved in ARF-mediated signalling in land plants

Our analysis suggests that during evolution, an ancestral RAV/ARF gene duplicated to give rise to ARF and RAV lineages. The ancRAV/ARF protein presumably harboured B3 and PB1 domains, and a BRD/LFG motif. Further, our data indicates that PHIP[cTudor] was coopted to become the structural ARF[DBD] chassis. As distantly related descendants of ancestral proteins that donated domains to ARFs, we asked if extant PHIP or RAV proteins share functions with ARF proteins. We focused on M. polymorpha genes, as both are single-copy genes in this species. MpPHIP (Mp2g00710/MpBromo4) is ubiquitously expressed in M. polymorpha gemmalings, irrespective of the conditions, while MpRAV (Mp2g22250) expression seems to be somewhat restricted from actively dividing cells as the apical notch and the sporophyte[22]. We generated CRISPR/Cas9 genome-edited mutants in both genes and obtained Mpphip[GE] alleles showing growth-impaired phenotypes, with small, sometimes aberrant, thallus formation, but typical apical notch architectures, and bearing gemmae-producing cups (Fig. 3a, Supplementary Fig. 7a–c). These plants respond normally to auxin and produced gametangiophores in response to far-red light as wild-type plants (Fig. 3b, c, Supplementary Fig. 7d). This suggests no genetic nor functional resemblance to either ARF1 or ARF3 function. Indeed, Mpphip[GE] mutants showed normal auxin-dependent MpWIP1 gene expression (Fig. 3d). While we failed to recover Mprav mutants in wild-type background, we could generate mutants in the Mparf1 background (Supplementary Fig. 8a, b). The Mprav genome-edited Mparf1 plants phenocopied Mparf1 (Fig. 3e, f, Supplementary Fig. 8c), showing unaltered gene expression in response to auxin (Fig. 3g). Thus, while PHIP and RAV genes are evolutionarily related to ARFs, they do not appear to be functionally connected. This suggests that the incorporation of ARF proteins in auxin response and auxin-dependent growth and development is a trait that emerged after departure from the ancestral gene copies.

## Identification of a deeply rooted ABC "pan-ARF" class, sister to all ARFs

Previous rooted-tree phylogenetic analyses of the ARF subfamily relied on low phylogenetic signal due to small domains, i.e., B3 and PB1s. Many sequences show a high degree of divergence, as for PB1's domain IV or B3s as those of ETTIN/ARF3 clade, leading to recovering conflicting topologies after tree building attributable to long-branch attraction[23]. The discovery of an evolutionary link between PHIP[cTudor] domain and ARF[DBD] prompted us to use the PHIP homology region as an outgroup to root extant ARF sequences. We therefore rooted our ARF tree using the PHIP[cTudor] clade as outgroup, and retrieved all known streptophyte ARF clades, AB, A, B, and C (Fig. 4a). In addition, we found a basal, fully supported, clade with Chlorokybophyceae and M. viridae sequences, considered to be part of the earliest diverging extant streptophyte lineages. This clade appears as a sister clade to all other ARF sequences, suggesting that these sequences neither belong to C nor AB clades, and instead belong to a different class, the ABC-ARFs. This data indicates that an ancestral ABC-like ARF duplicated during streptophyte evolution into the C- and AB-ARF classes. ABC-class ARFs share discrete homology with both AB- and C-classes and likely hold ABC-specific features. This is in line with the behaviour in the clustering of both MvARF/CmARF[B3]s and CmARF[PB1] which neither fell within C-ARFs nor AB-ARFs clades (Fig. 2b, f).

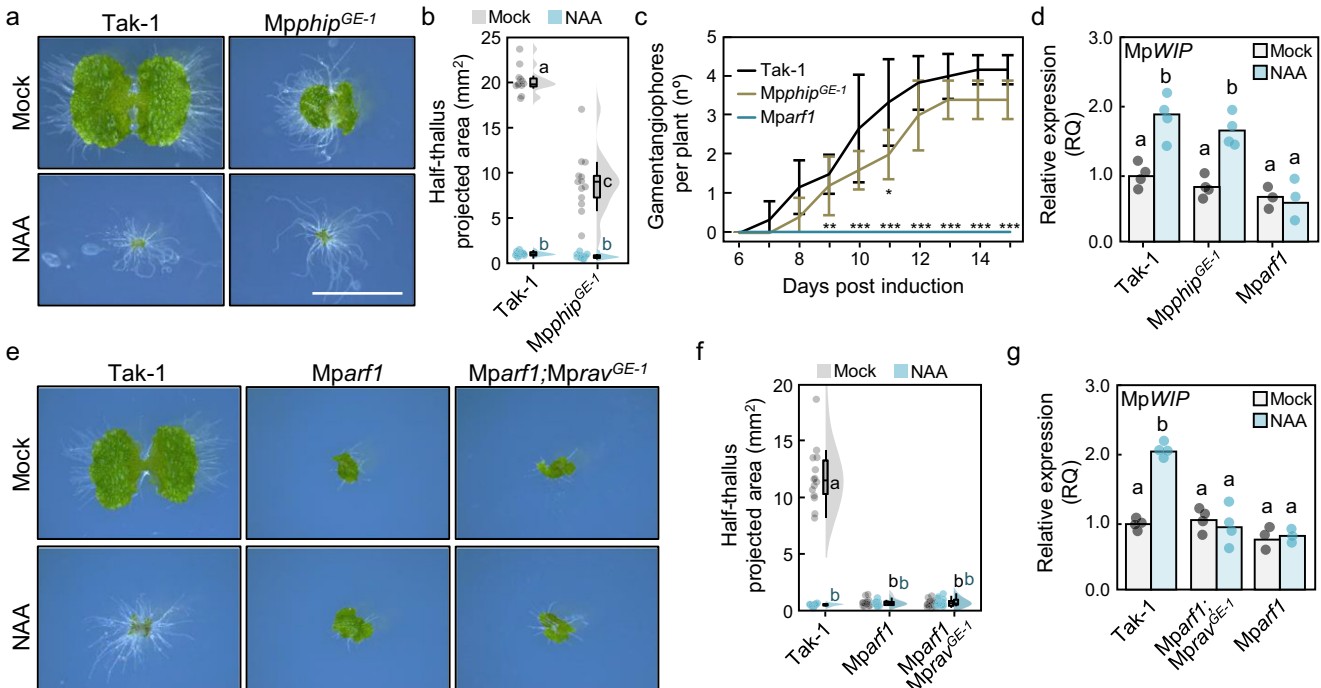

**Fig. 3 | PHIP and RAV are not related to ARF function in *M. polymorpha*.**
**a** Pictures of 10-day-old wild-type (Tak-1) and genomic edited *phip* plants grown in mock (DMSO) or auxin (3 μM NAA). Scale bar, 5 mm. **b** Raincloud plot of thallus area measurements (in halves) of 10-day-old wild-type and *phip* mutant plants grown in (**a**). n = 12,14,11,11 (left to right, mock/treatment). Statistical groups are determined by Tukey's Post-Hoc test (*p* < 0.05) following one-way ANOVA. **c** Quantification of gametangiophore formation after sexual organ induction (continuous far-red light, cFR) in 10-day-old plants. Data represents mean and error bars each day after transfer to cFR, and standard deviation of five plants. Asterisks indicate statistical difference compared to the wild-type group (Tak-1) by Kruskal–Wallis test (**p* < 0.05; ***p* < 0.01; ****p* < 0.001). **d, g** Expression analysis of Mp*WIP* by qRT-PCR in 10-day-old plants treated for 1 h with 3 μM NAA or DMSO (Mock), using Mp*SAND*

and Mp*EF1α* as reference genes. Dots represent average of two technical replicates of biological replicates (n = 4), but for Mp*arf1* (n = 3); bars represent the average of the biological replicates. Statistical groups are determined by Tukey's Post-Hoc test (*p* < 0.05) following one-way ANOVA. **e** Pictures of 10-day-old wild-type (Tak-1) and genomic edited Mp*arf1;rav* plants grown in mock (DMSO) or auxin (3 μM NAA). Scale bar, 5 mm. **f** Raincloud plot of thallus area measurements (in halves) of 10-day-old wild-type and *rav* mutant plants grown in (**e**). n = 14. Letters (black, mock; blue, NAA) indicate statistical groups as determined by Tukey's Post-Hoc test (*p* < 0.05) following one-way ANOVA. Source data are provided as a Source Data file. Boxplots in Raincloud indicate the following parameters: centrum, median; upper bound, first quartile; lower bound, third quartile; whiskers maximum and minimum refer to highest and lowest values, respectively, within 1.5*inter-quartile range (IQR).

ARFs are present in all streptophyte algae lineages. However, AB- and C-classes have been identified sparsely, and in a seemingly mutually exclusive pattern, with the only exception of some Coleochaetophyceae species[10,11]. We searched for ARF sequences in up-to-date streptophyte databases, transcriptomes, and new genome assemblies and annotations (Supplementary Data 3). We confirmed that A- and B-classes are strictly found in land plants, and found that the AB-class is more prevalent among streptophyte algae than previous data showed (Fig. 4b). This allowed us to resolve the previously observed incoherent mutual exclusion of AB- and C-class ARFs in algae, as we found that most species harbour both AB- and C-class ARFs. This likely occurred due to unannotated genes, with notable exceptions such as the lack of AB-class in the Charophyceae class, or the C-class within the Desmidiales order (Class Zygnematophyceae). More importantly, we found that AB- and C-class origin predates Klebsormidophyceae divergence from the stem streptophytes, likely after an ancestral ABC-ARF duplication. Our data entails an updated and exhaustive view of ARF evolutionary history and indicates that C-class predates A- and B-classes. The finding of ABC-class suggests that C-class is unlikely to be reminiscent of the ancestral ARF function.

## A-class ARFs originated in land plants after the neofunctionalization of transcriptional activity

To investigate the functional conservation between ARF classes and infer ancestral functions within each class, we performed Mp*arf* complementation assays with a subset of algal ARFs from different clades and species. We chose a set of representative ABC- (CmARF),

AB- (SmARFs and PmARFs), and C-ARFs (MeARFc and SpARFc) from annotated genomes with full-length architectures (DBD to PB1). We first introduced full-length coding sequences (CDS) under the control of the respective endogenous ARF promoters (*pro*Mp*ARF1*) in the A-class Mp*arf1-4* mutant background[14]. No algal CDSs complemented the A-class Mp*arf1* mutant in terms of thallus area or auxin response, similar to what has been shown for MpARF2 and MpARF3, and contrasting with the full complementation when using Mp*ARF1* coding sequence (Fig. 5a, b, Supplementary Fig. 9a). Interestingly, mild thallus area recovering occurred by introducing the ABC-class ARF from *Chlorokybus melkonianii* (CmARF). However, many of these lines produced plants presenting developmental defects as undifferentiated outgrowths or lack of discernible organs. Similar defects were found with *Mesotaenium endlicherianum* C-ARF lines, which produced gemma cup malformations and retarded growth even compared to Mp*arf1* plants (Supplementary Fig. 9b, c). This suggests that ABC- and C-classes induce pleiotropic phenotypes when expressed at moderate levels in Mp*arf1*. Since no functional conservation seems to exist between ABC-, AB-, or C-class algal ARFs and A-class ARFs, we aimed to confirm whether A-class functionality is conserved among land plants. To avoid lineage-specific sub- and neofunctionalization effects, we chose the single A-class ARFs from the moss *Takakia lepidozioides* and the hornwort *Anthoceros agrestis*, representing lineages that diverged from liverworts around 450 and 420 million years ago, respectively[24]. Both ARFs complemented Mp*arf1* mutant, indicating that A-ARF function is exclusive and likely represents a synapomorphy of the class (Fig. 5a, Supplementary Fig. 9a, d).

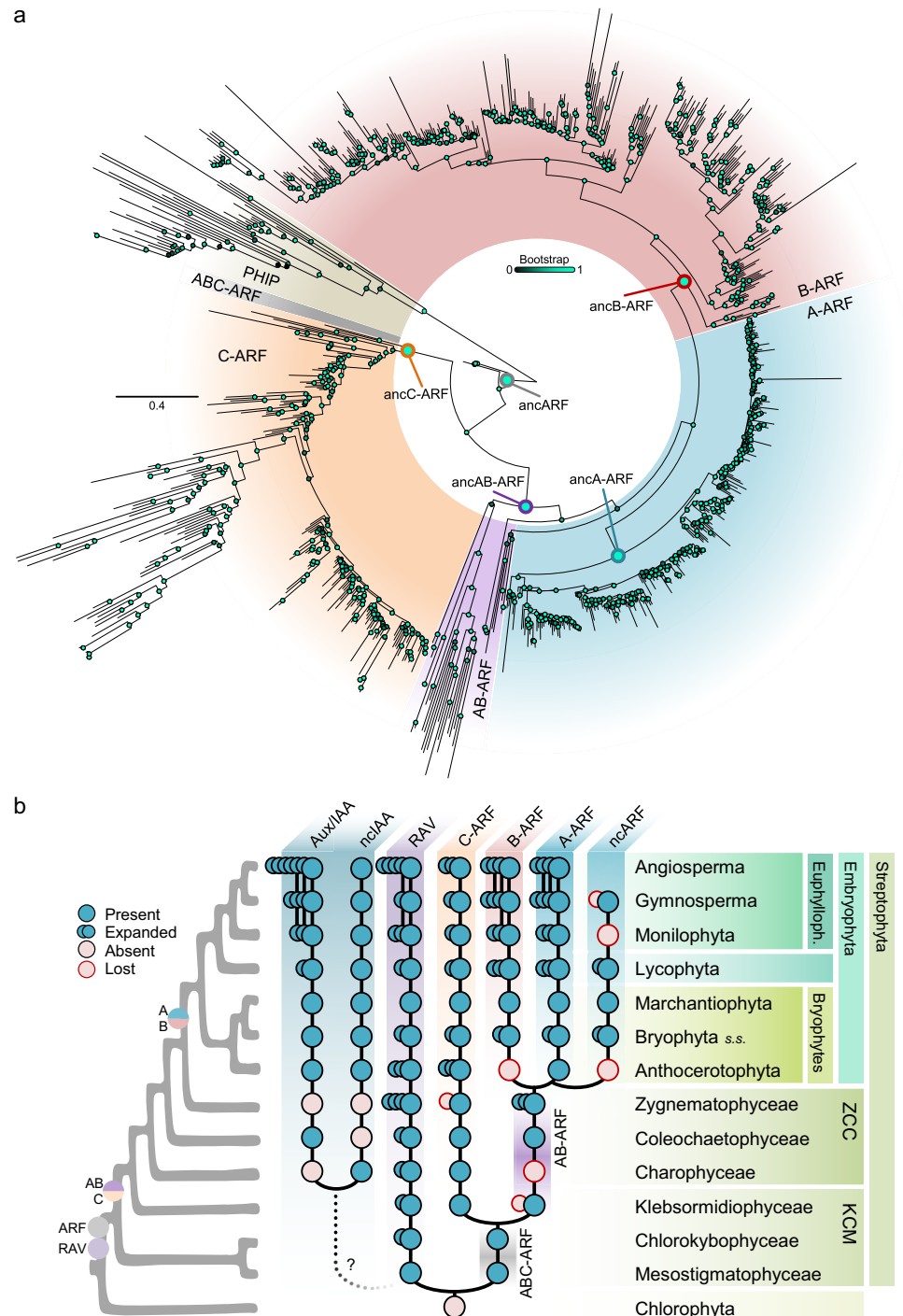

**Fig. 4 | ARF originated and diverged from a single ABC-class. a** Phylogenetic tree of ARF proteins using DD-to-AD protein sequences, rooted using PHIP sequences. Bootstrap values are indicated as color-coded bubbles in branch nodes, with the main ARF class ancestral nodes highlighted. Scale bar represents distance in substitutions per residue. **b** Reconstruction of the evolutionary pathway of ARF and related proteins in streptophytes as a schematic summary of presence/absence extracted from data in phylogenetic analyses. Right tree depicts the known phylogenomic tree of plant lineages with ARF major origin and duplication events marked. Expansion indicates lineage-specific family expansions inferred from phylogenetic data as a qualitative approximation. Source data are provided as a Source Data file.

The molecular features commonly attributed to A-class ARFs are i) transcriptional activation, and ii) specific interaction with the Aux/IAA co-repressors, thus being subjected to auxin-control. We assessed if these features represent class-specific neofunctionalized traits, or if these are also present in other algal classes. For the former, we performed transcriptional activation assays in a yeast synthetic system with several full-length ARF proteins (Fig. 5c). All tested A-class ARFs

showed clear induction of the reporter, suggesting that A-class ARFs act as strong transcriptional activators, and consistent with the identification of small regions that can activate gene expression in a heterologous assay[25]. The remaining ARFs were unable to induce transcription but for a mild ability of *Spirogyra pratensis* (Zygnematophyceae) C-class ARF activation. We confirmed this in a subset of these ARFs using an Arabidopsis protoplasts assay (Fig. 5d). Our

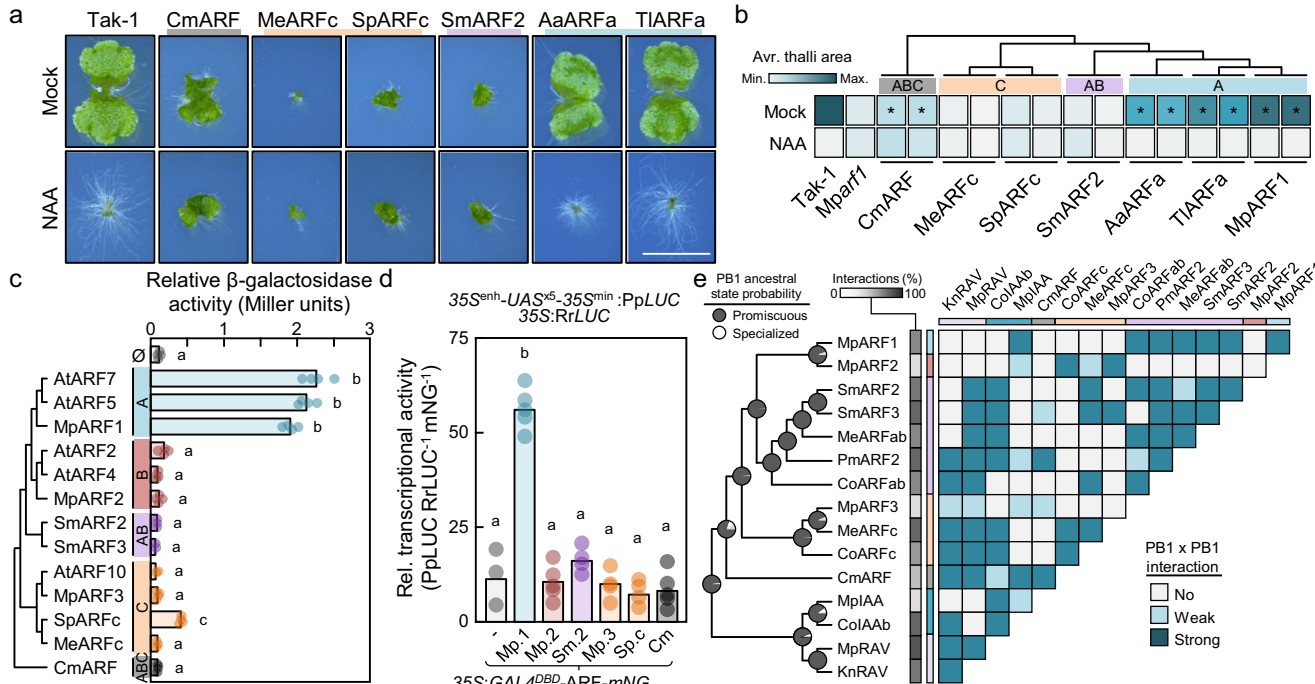

**Fig. 5 | A-class ARFs evolved through fast neofunctionalization. a** Pictures of 10-day-old plants grown in mock (DMSO) or auxin (3 μM NAA) in a Mp*arf1* complementation assay. Full length CDS of referred ARFs are expressed under the endogenous MpARF1 promoter in a Mp*arf1* background. Scale bar, 5 mm. **b** Heatmap representation of Mp*arf1* thallus phenotype complementation with full length ARFs of 10-day-old plants grown in mock (DMSO) or auxin (3 μM NAA) extracted from thallus area measurements. Asterisks indicate statistical differences with Mp*arf1* as determined by Tukey's Post-Hoc test (*p* < 0.05) following one-way ANOVA (see Supplementary Fig. 9). Asterisks indicate statistical difference compared to the mutant in mock conditions. Upper tree indicates phylogenetic representations among the ARFs used in the complementation assay. **c**, Yeast transactivation assays of full length ARFs fused to the yeast Gal4 DNA binding domain showing the quantification of the *UAS:LacZ* reporter activation as colorimetric-measured β-galactosidase activity. Dots represent the average of three

semi-technical replicates in independent biological replicates (independently transformed lines, *n* = 4). **d** Dual luciferase transactivation assay in Arabidopsis protoplasts using a *LUC* gene under the control of 5xGal UAS motif as reporter, *35S:REN* as ratiometric control, and the yeast Gal4 DNA binding domain fused to ARFs as effectors, fused additionally to mNeonGreen for protein expression normalization. **e** Qualitative summary of interspecies PB1 pairwise interaction assays obtained from yeast-two-hybrid quantitative β-galactosidase activity assays (extracted from Supplementary Data 2). In (**c**, **d**), statistical groups are determined by Tukey's Post-Hoc test (*p* < 0.05) following one-way ANOVA. Cm *Chlorokybus atmophyticus*, Me *Mesotaenium endlicherianum*, Sp *Spirogyra pratensis*, Sm *Spirogloea muscicola*, Pm *Penium margaritaceum*, Co *Coleochaete orbicularis*, Kn *Klebsormidium nitens*, Aa *Anthoceros agrestis*, Tl *Takakia lepidozioides*. Source data are provided as a Source Data file.

protoplast assay showed no activation function of SpARFc, suggesting that the mild activation in yeast is not its native function. Our findings support the idea of ARF transcriptional activation being exclusive to A-class ARFs.

To understand the auxin-regulation (Aux/IAA interaction) and oligomerization potential (homotypic interaction) of the ARF family, we explored the interaction profile of an expanded set of PB1s from algal and land plant ARFs, Aux/IAA, and RAVs (Fig. 5e, Supplementary Fig. 10). We observed several positive interactions between heterologous algal ARF[PB1]s and Aux/IAA[PB1]s, and a widespread homotypic interaction profile. Similarly, interactions with RAV[PB1]s are common among ABC, AB, and C-ARF[PB1]s, ranging from 47% to 66% capacity to interact with the rest of PB1s (percentage of the total possible PB1-PB1 interactions in the matrix). In contrast, land plant ARFs and Aux/IAA showed a higher restriction in their interaction potential, with both MpARF2 and MpARF3 having the lowest interaction capacity (13%), followed by MpIAA (33%). Interestingly, the A-class MpARF1 was able to establish interaction with most AB-class ARFs, indicating that B-class ARFs likely restricted their PB1 interacting potential during land plant evolution from a more promiscuous PB1 ancestor. Thus, ancestral state inference strongly suggests that specialization is a derived trait appearing independently in different taxa and gene families. Altogether, this data suggests that the main trait acquired during the emergence of the A-class was transcriptional activation from a transcriptional repressor ancestor.

## C-ARF function is conserved and distinct from ABC-ARF function

To assess the conservation of C-class function, we introduced a similar set of algal ARFs into the Mp*arf3-1* mutant[11]. We included the A- and B-class MpARF1 and MpARF2. Due to Mp*arf3* mutants not producing gemmae, we evaluated the ability to complement by measuring apical growth after cutting. None of the non-C classes could complement Mp*arf3* phenotype to any visible degree (Fig. 6a, b, Supplementary Fig. 11a, b). However, the algal C-class MeARFc showed comparable complementation levels to that of MpARF3 coding sequence, as shown by the full recovery of the gemma formation phenotype characteristic of the Mp*arf3* mutant (Fig. 6c, Supplementary Fig. 11a, c), followed by SpARFc to a much lesser extent. Similar complementation has been found using Arabidopsis C-class ARFs to complement the mutant, in which thallus shape and gemma formation were restored, while the inhibited regeneration of Mp*arf3* was not complemented[15]. This suggests a deep conservation in the main functionalities within the C-class during hundreds of millions of years, but also the existence of lineage-specific acquisition of additional functions. CmARF expression in Mp*arf3* did not complement the mutant phenotype and instead produced pleiotropic effects, similar to what occurs in a Mp*arf1* background (Supplementary Fig. 11d). Our genetic analysis aligns with our phylogenetic data and indicates that ABC-ARFs cannot complement either Mp*arf3* or Mp*arf1*, in line with ABC-class ARFs having a different functionality from either A or C-classes.

Our data shows that full-length coding sequences of different ARF class cannot complement other ARF class mutants, indicating deep class-specific specialization. Due to the lack of viable B-class mutants, it remains to be explored whether B-class function can be complemented by AB-class or both A- and B-class divergence was followed by independent neofunctionalization of each class.

## Domain specialization underlies ARF class diversification

The functionality of each ARF domain is independent and separable from one another[14]. To dissect if different protein domains were the main drivers of sub- and neofunctionalization within the ARF family, we performed domain swap assays of different algal ARFs into our genetic models. The DBD of A- and B-classes are closely related and have been suggested to bind the same cis-binding elements. We first performed Mp*arf1* complementation assays in which we introduced MpARF1 versions where we had substituted the DBD for different algal ARF^DBDs. We found that only the chimeras with AB-class DBDs from *S. muscicola* and *M. endlicherianum* were partially able to complement the Mp*arf1* mutant (Fig. 7a, b, Supplementary Fig. 12a), resembling swaps using the B-class MpARF2^DBD[14]. This indicates that AB^DBD functionality is similar to that of A- and B-classes, but not C, regardless of the lineage. The AB-class PmARF1^DBD was unable to complement, indicating that neofunctionalization has occurred in some algal AB-class ARFs, however, as expected for A- and B-classes, it is able to form homodimers in solution, as assessed by SEC-MALLS (Supplementary Fig. 12b). This suggests that AB-ARFs are likely able to bind the same sequences as A- and B-ARFs. We used electrophoretic mobility shift assays to confirm that AB-ARFs are indeed able to bind a known bipartite auxin response elements (AuxRE) in a cooperative and concentration-dependent manner (Fig. 7c, Supplementary Fig. 12c–e), resembling A- and B-class ARF properties.

Our results align with the notion that C-class DBD function is distinct from A- and B-class. This suggests that, in contrast to A and B divergence, AB and C divergence was governed by DBD specialization. To test this hypothesis, we generated a series of Marchantia ARF domain swaps between DBD, MR and PB1 domains and introduced these into the Mp*arf3* mutant (Fig. 7d, Supplementary Fig. 13a). ARF chimeras bearing MpARF3^DBD were able to complement the mutant phenotype to different levels. In contrast, both MpARF1^DBD and MpARF2^DBD swaps were incapable of rescuing known Mp*arf3* mutant phenotypes. This supports the DBD as the major functional difference between C-class and A/B-classes. Interestingly, while MR and PB1 swap chimeras of either A- and B- classes rescued signature Mp*arf3* phenotypes as the flat thalli and gemma formation, they also produced abnormal plants, suggesting the existence of C-class specific functions encoded in the MR (Fig. 7d, Supplementary Fig. 13a, b). Our results suggest that DBD function is the main difference between C-class and the rest of the classes. To understand the level of conservation of this function, we introduced algal ARF^DBDs from ABC- (CmARF), AB- (SmARF) and C-classes (MeARFc) into Mp*arf3* (Fig. 7e). As expected from the complementation attained by the full-length version, *M. endlicherianum* C-class DBD rescued Mp*arf3* mutant phenotype, but neither AB- nor ABC-class DBDs were able to complement the mutant. Coinciding, a MpARF3 chimera with *M. endlicherianum* C-class MR-PB1 was also fully able to complement the mutant phenotype (Fig. 7f, Supplementary Fig. 13c). This suggests that the C-class DBD neofunctionalized soon after diverging from the AB-class, and indicates that this was the main region of functional specialization within this class, and that it has retained most of its original function during evolution.

## Discussion

Benefiting from the increasing amount of genomic data and protein homology searches, we present here the most complete picture to date of the origin and evolution of the major regulators of auxin

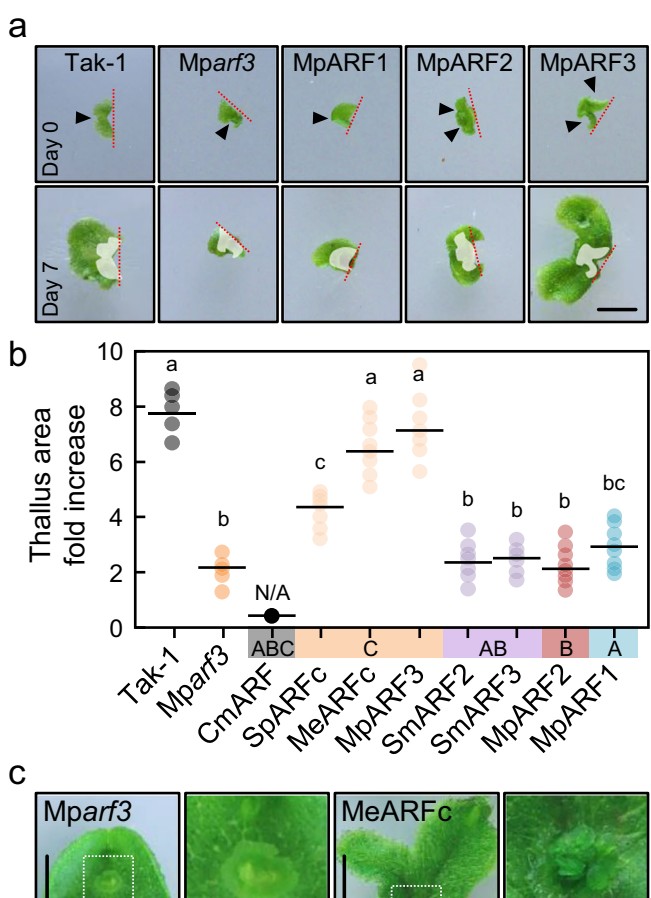

**Fig. 6 | C-class ARFs function is specific and deeply conserved. a** Pictures of Mp*arf3* mutant complemented with different ARF classes. MpARF1, MpARF2, and MpARF3 coding sequences are expressed under the control of the endogenous Mp*ARF3* promoter in the Mp*arf3* mutant. Upper row represents 20-day-old apical notches right after excision from adult plants. Lower row are the same plants after seven days of re-growth. White-shaded area in lower row is the same area occupied at day 0. Red dotted line indicates excision, while filled arrowheads point to apical notches present at excision. Scale bar, 5 mm. **b** Dot plot of notch-driven thallus growth measured as projected area fold-change 14 days after excision (percentage of times day 0 area). ARF coding sequences are expressed under the control of the endogenous Mp*ARF3* promoter in the Mp*arf3* mutant. Mp*arf3* plants expressing CmARF do not produce visible notches, preventing comparable excision. Dots represent thallus area growth percentage average of three notches derived from a single plant; *n* = 5,5,6,8,7,7,6,8,8 (left to right). Statistical groups are determined by non-pooled Welch's *t*-test and Benjamini–Hochberg adjustment (adjusted *p* < 0.01 for non-overlapping letters). **c** Pictures of regenerated plants showing the phenotype of a MeARFc-complemented Mp*arf3*. White dotted squares in first and third images indicate the zoomed are shown in the second and fourth pictures, respectively, highlighting a mature gemma cup. Cm *Chlorokybus melkonianii*, Me *Mesotaenium endlicherianum*, Sp *Spirogyra pratensis*, Sm *Spirogloea muscicola*. Source data are provided as a Source Data file.

transcriptional responses, the ARFs. We uncover the origin of their DBD scaffold in a eukaryotic family of chromatin reader proteins, PHIP/BRWD, pinpointing the relevance of the acquisition of pre-existing domains through shuffling during gene neofunctionalization (Fig. 8a). This allows us to propose a plausible hypothesis for the origin of ARFs by the complex fusion of a partial *PHIP* gene and a *RAV/ARF* duplicated gene in a streptophyte common ancestor (Fig. 8b). It remains unknown what was the history of *RAV-ARF* gene split, especially regarding the timing of *PHIP* fragment recombination. It is plausible that an ancestral *RAV/ARF* gene bearing at least B3,

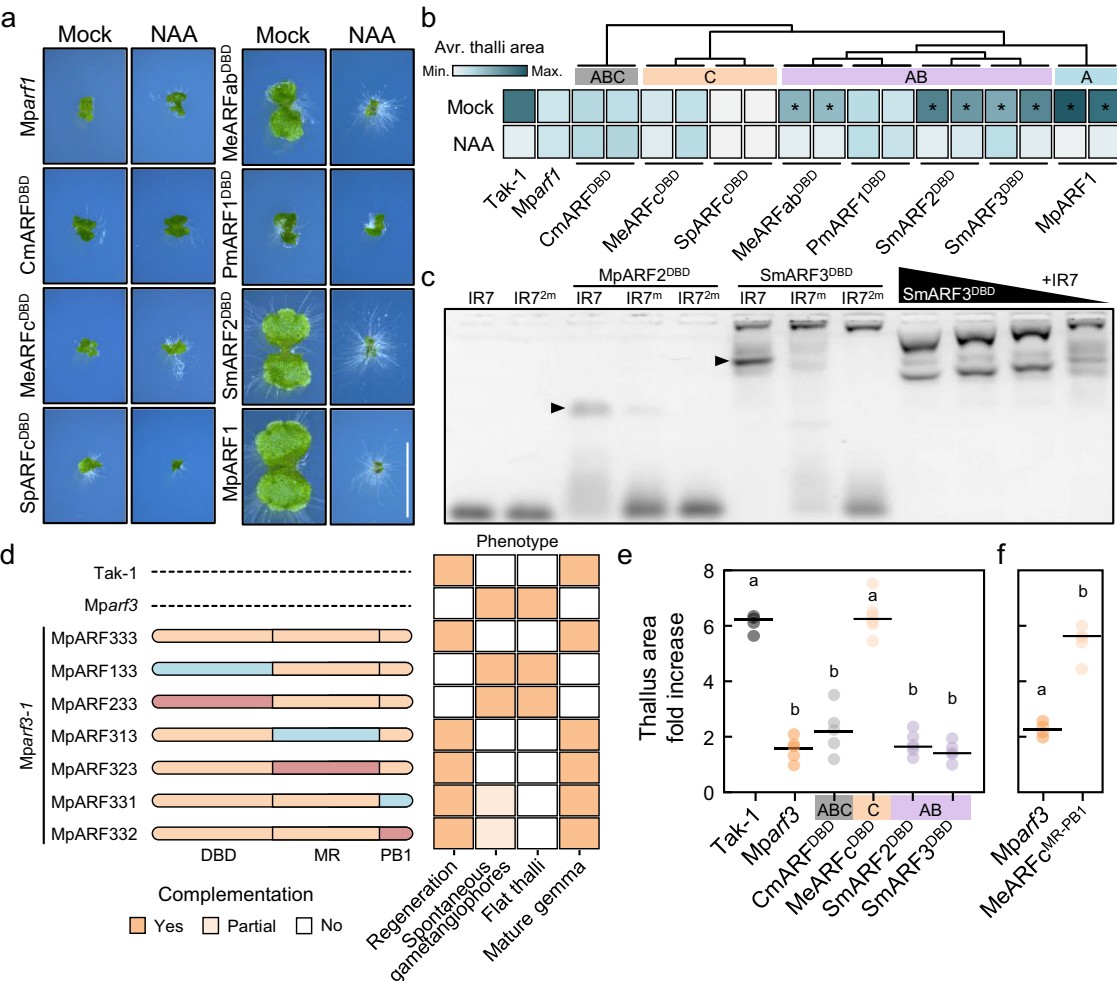

**Fig. 7 | ARF classes diverged through sequential domain specialization.**
**a** Pictures of 10-day-old plants grown in mock (DMSO) or auxin (3 µM NAA) in a Mp*arf1* complementation assay. Chimeric MpARF1 with heterologous ARF^DBD sequences as indicated are expressed under the endogenous MpARF1 promoter in a Mp*arf1* background. Scale bar, 5 mm. **b** Heatmap of Mp*arf1* thallus phenotype complementation with MpARF1-ARF^DBD chimeras of 10-day-old plants grown in mock (DMSO) or auxin (3 µM NAA) extracted from thallus area measurements. Asterisks indicate statistical differences with Mp*arf1* as determined by Tukey's Post-Hoc test ($p < 0.05$) following one-way ANOVA (see Supplementary Fig. 12). **c** Electrophoretic mobility assay showing MBP-SmARF3 DNA-binding domain interaction with a bipartite ARF-specific binding site (IR7, inverted repeat, 7 bp spacing between two Auxin Response Elements, AuxRE), with one or two mutated AuxRE (IR7^m and IR7^2m, respectively). Marchantia B-class ARF (MpARF2) DBD is shown as positive control. Right-most lanes show a MBP-SmARF3 protein titration experiment indicating concentration-dependent complex-formation ([SmARF3^DBD]=100,

50, 25, 5 µM, left to right). This assay has been performed twice with equivalent results. **d** Summary MpARF3 domain swaps with MpARF1 and MpARF2 in Mp*arf3*-related phenotypes complementation extracted from experiments shown in Supplementary Fig. 13. Full complementation (*Yes*) is assigned when phenotype is comparable to that of the wild-type (Tak-1) instead of Mp*arf3*. *Partial* complementation indicates a quasi-wild-type phenotype. **e, f** Dot plots of notch-driven thallus growth measured as projected area doubling 14 days after excision (percentage of times day 0 area). Domain-swapped MpARF3 chimeras are expressed under the control of the endogenous Mp*ARF3* promoter in the Mp*arf3* mutant. Dots represent thallus area growth percentage average of three notches derived from a single plant. **e** MpARF3-ARF^DBD chimeras; $n = 5$. **f** MpARF3-MeARFc^MR-PB1 chimera; $n = 4$. Statistical groups are determined by non-pooled Welch's *t*-test and Benjamini–Hochberg adjustment (adjusted $p < 0.01$ for non-overlapping letters). Sm *Spirogloea muscicola*, Pm *Penium margaritaceum*. Source data are provided as a Source Data file.

BRD/LFG and PB1 domains, eventually underwent gene duplication giving rise to both gene lineages. At some point, before or after this duplication, a cTudor domain recombined (and was either afterwards lost in the RAV lineage, or acquired in the ARF lineage), giving rise to the extant ARF protein architecture. PHIP^cTudor domains are known to bind methylated H3K4[26], consistent with their relatedness to other Tudor-like domains. However, the homologous domain in ARFs no longer carries this function, or at least histone binding is not functionally relevant under our experimental conditions. Instead, the presence of the B3 domain seems to direct the main function of the domain, DNA-binding. RAVs and ARFs are closely related and share a common origin, yet show very different B3 specificities for DNA, i.e.: CNCCTG and TGTCNN motifs, respectively[27]. This coincides with the divergent branches observed in both B3 clades (Fig. 2b). It remains

unclear how both binding preferences diverged from an ancestral RAV/ARF B3 and multiple scenarios are plausible. However, these evolutionary details are not resolvable with current technologies, and the exact events might remain a mystery. The evolutionary story after the RAV-ARF split was likely characterized by an early and fast evolution of DNA-binding preferences within each subfamily B3, and an initially slow divergence of the PB1, which eventually led to the origin of the nuclear auxin pathway.

The updated ARF phylogeny presented here fills gaps that previously led to misinterpretations (Figs. 4b, and 8c). Previous studies assigned an ABC-ARF to the C-class based on limited sequence availability and DNA-binding properties[10,11]. Here, we unambiguously locate Chlorokybophyceae and Mesostigmatophyceae ARFs to this novel ABC-class clade, sister to the remaining ARFs. We also revisit

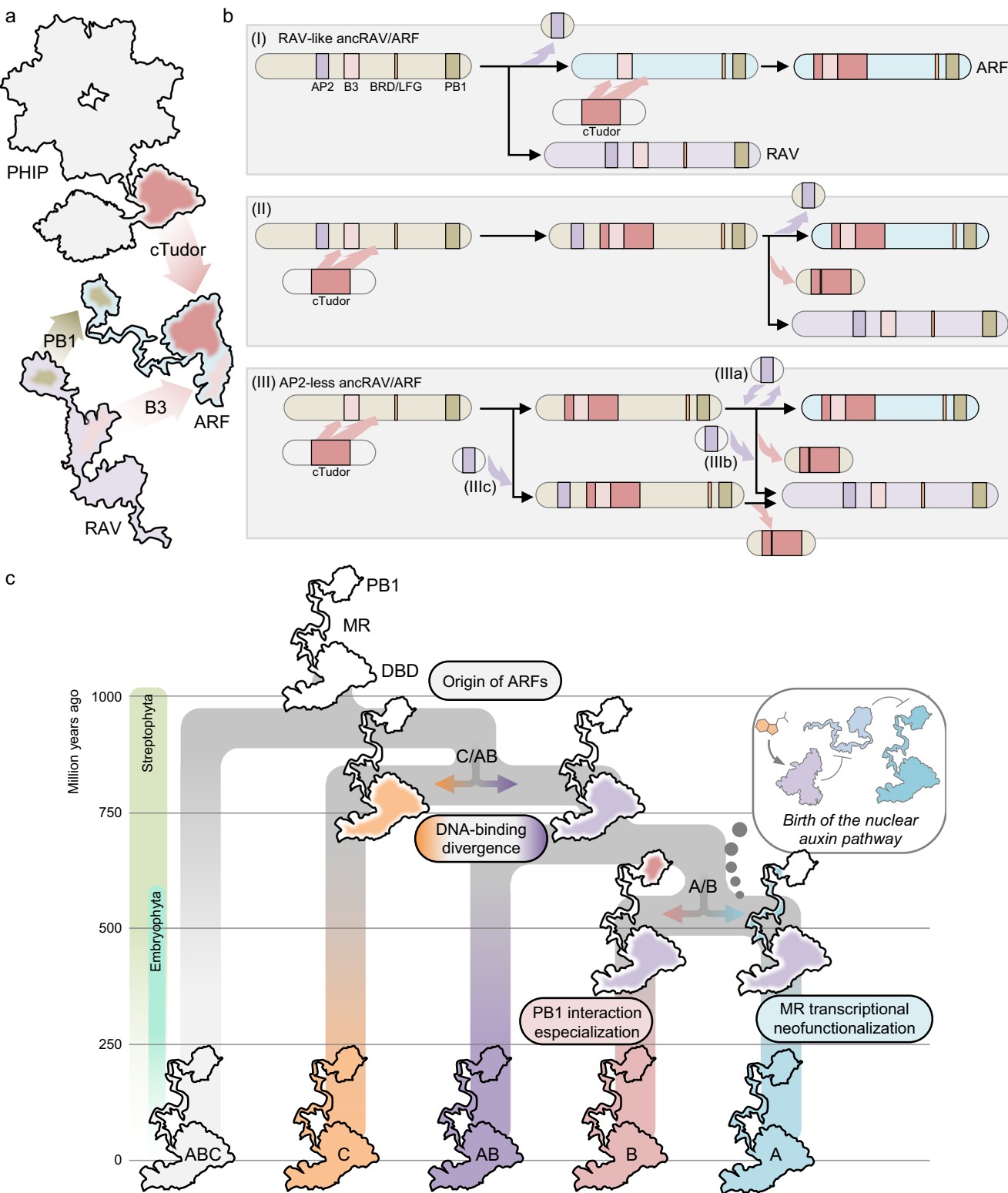

**Fig. 8 | Origin and evolutionary history of ARF proteins. a** Homologous domain origin leading to ARF proteins. Cartoon adaptation from known structures of domains, connected by an artistic representation of regions without a known structure. ARF homologous domains are highlighted in the related protein cartoons (Dark rosy-brown shade, cTudor from PHIP; Tan-brown, PB1, and light rosy-brown, B3 domains from RAV). **b** Hypothetical step-wise domain acquisition originating ARF and RAV domain architectures from an ancestral ARF/RAV gene. (I) and (II) assume a RAV-like ancestral RAV/ARF (AP2 + B3 containing). (I) Duplication of ancestral copy giving rise to ARF and RAV lineages; ARF-independent acquisition of PHIP-cTudor scaffold. (II) Acquisition of cTudor in ancARF/RAV and subsequent duplication and loss of this domain in the RAV lineage. (III) Roadmaps from a non-AP2 containing ancRAV/ARF; subsequently AP2 domain acquired or lost at different points. BRD stands for B3 repression domain. **c** ARF evolution from the ancestral ARF. Major divergence points associated to molecular sub- and neofunctionalization events are indicated.

the origin of the AB-class, considered previously present only in Zygnematophyceae and Coleochaetophyceae. Our data shows a prevalent presence of AB-ARFs in different Klebsormidiophyceae species, pushing its origin 300 million years back, and exemplifying the difficulty of relying on a single species for lineage-wide evolutionary analyses. In contrast, AB-ARFs are absent in available Charophyceae algal databases, but the generation of new genomic resources could change this view. Here, we have also showcased the strength of heterologous complementation analyses, which confirm previous hypotheses about the divergence between C-ARF DBD and the other classes. These assays are consistent with an ARF evolutionary history of two major duplication events targeting different domains (Fig. 8c). This first of these duplications would involve the ABC-ARF gene of a common ancestor of Klebsormidiophyceae and Phragmoplastophyta, and gave rise to the AB and C orthologous lineages. The main functional divergence of these orthologs occurred through fast changes in their DBD sequences, which have subsequently become fixed and remained deeply conserved for both lineages. A second major event occurred in the AB-ARF gene of a land plant common ancestor, and gave rise to the A and B lineages. This event was followed by the subsequent evolution of two essential traits for the rise of the nuclear auxin signalling pathway: i) the gain of transcriptional activation capacities in the A-ARFs, and ii) the specialization and restriction of the oligomerization capacities of PB1s, likely through restriction of its interaction promiscuity in B-ARF, leaving the DBD functions of both A- and B-ARFs practically unaltered.

Given their phylogenetic position, none of these duplications can be attributed to inferred whole genome duplication events[28], suggesting that small-scale duplications underlie these major ARF branching points. The first event coincides with the possible origin of ancestral filamentous multicellularity, one billion years ago[29], which may have endowed streptophytes with the ability to colonize many subaerial habitats prior to land plant emergence. It is thus plausible that these filamentous ancestors benefited from acquiring two divergent ARFs to regulate different gene networks in order to adapt to different ecological niches. The second event occurred close to the origin of embryophytes, almost undoubtedly in an ancestor of all land plants, together with hundreds of other gene families concurrently appearing and expanding[24]. While the initial physiological implications of the duplication generating A- and B-ARFs is unknown, both lineages have been selected and maintained in most land plants, with few exceptions (as the absence of B-class in hornworts), underlining their functional importance. This also indicates that the antagonism between these two classes was acquired very soon during early land plant evolution, likely concomitantly to their incorporation into the nuclear auxin pathway. Our analysis indicates that the land plant common ancestor harboured a single gene for each ARF class (A, B, but also C), which aligns well with previous studies indicating that a single A- and B-ARFs are sufficient to exert a wide range of auxin-dependent transcriptional responses[9,11,14]. In fact, our data indicates that both bryophyte and tracheophyte common ancestors also contained this minimal set of ARFs, suggesting that, aside from lineage-specific duplications, the first major ARF duplications occurred only in the euphyllophyte ancestor (*ca.* 420 mya).

This scenario also implies the rapid acquisition of a functional TIR1-Aux/IAA auxin-dependent regulatory module in an embryophyte common ancestor. Aux/IAA have been found patchily in only two lineages of streptophyte algae, and it remains unclear when and how these appeared. The earliest diverging lineage from the stem Streptophyta containing Aux/IAA genes is the Charophyceae[30], but these sequences belong to the non-canonical clade of Aux/IAA. The other lineage containing Aux/IAA genes is the Coleochaetophyceae[15], whose sequences seem to fall within the clade of canonical Aux/IAA, hindering ancestral status prediction. Our PB1 phylogenetic analysis suggests that Aux/IAA could have evolved from a fragmented duplication of a RAV gene (Fig. 2f), instead of an ARF. The long branches at the base of these clades however indicate these are fast evolving clades and the inferred tree topology may be a long branch attraction-derived error. Thus, while predating land plant origin, the emergence of Aux/IAA remains unclear. It is also unknown when Aux/IAA acquired the auxin-dependent degron, although our PB1 phylogeny supports this occurring after the duplication of the non-canonical and canonical clades, as previously proposed[15], instead of non-canonical Aux/IAA having evolved after secondarily losing the DII motif.

This study identifies the birth of the ARF subfamily in domain shuffling events, and a series of subsequent gene duplication events followed by neofunctionalization as essential evolutionary mechanisms during the assembly of new hormone signalling pathways.

## Methods
### Sequence identification
Identification of distant ARF homologous sequences was performed using the phmmer search of HMMER 3.4 (http://hmmer.org/) and standard BLASTP. DBD protein sequences from AtARF1, MpARF1 and CmARF were set as phmmer queries against the Reference Proteomes database (The Uniprot Consortium, 2019) without Taxonomy restriction using default cut-offs. Lists of domain architecture were manually merged into similar compositions discarding single domain hits. In parallel, AD protein sequences of the same ARFs were used in BLAST searches in additional eukaryotic proteomes. Taxa sampling was extended to entire lineage proteomes in NCBI, UniProt, Phytozome, Mycocosm, Phycocosm, and oneKP databases to confirm the absence or presence of specific proteins or domains when no hits were initially found. To avoid ARF hits concealing PHIP-like hits, human PHIP sequence was used as query in streptophytes. To identify ARF sequences in an extended set of streptophyte algae databases, we performed BLASTP searches with an e-value cut-off of 10E-10. All sequences were manually curated to discard incomplete or fragmented hits and to merge duplicates or highly similar protein ( > 95% sequence identity). All databases and genomic sources listed can be found in Supplementary Data 3.

### Sequence alignment and phylogenetic analyses
Different alignments were performed for full length proteins. In short, curated sequences were used to compute alignments in M-Coffee[31] by combining the pairwise alignment methods of MAFFT and ClustalW, and the multiple alignment methods MAFFT, T-Coffee, ClustalW, and POA2. Manual curation was followed by automatic trimming to discard poorly aligned regions (more than 80% gaps in position) using trimAl[32]. For specific domains, these regions were initially extracted from the full length raw alignment and then curated manually avoiding trimAl. To construct maximum likelihood phylogenetic trees, we first run the ModelFinder implementation of IQ-TREE[33] on clean alignments to predict the best substitution model based on the Akaike-and Bayesian Information Criterion. When AIC and BIC were incongruent, we used the minimal BIC score. IQ-TREE was run using the clean alignments and the predicted substitution model using 1000 Ultrafast bootstrap and the SH-like approximate Likelihood Ratio Test to infer branch statistic support.

Ancestral trait inference of PB1 interaction propensity at predicted nodes was performed using the Bayesian Binary MCMC (BBM) method implemented in RASP 4[34]. An input tree of PB1 sequences was built following the above indicated steps with IQ-TREE, and topology manually curated. Qualitative assessment of interaction propensity based in the interaction matrix was done by establishing a threshold of equal or above 50% positive interactions for promiscuous PB1, and specialized if below. BBM was then run using the standard Markov Chain Monte Carlo analysis parameters but for the number of cycles = 100,000.

## Protein structure prediction

Structures for PHIP and ARF were predicted using AlphaFold2 (AF2), for either full length proteins or specific domains with default settings[35]. Performance of AF2 for de novo ARF structure prediction was manually evaluated. Predicted full-lengths structures were trimmed to leave ARF$^{DD-AD}$ and PHIP$^{cTudor}$ domains. Only α-helices and β-sheets were used to align the structures and calculate RMSD. Visualization and figures were made in PyMOL (3.0.3., Schrödinger) and alignments made using the *align* command with 5 cycles of outlier rejection.

## Plant material and growth conditions

The Arabidopsis *arf5/monopteros* mutant alleles *mpB4149* and *mp-12* (SALK_149553) in the Col-0 background had been previously described[36,37]. Seeds were surface-sterilized with a bleach/ethanol (25%/75%) solution for 10 minutes followed by a 96% ethanol wash, and sown after drying on half-strength MS (Duchefa) pH 5.7 plates containing 0.8% agar. Stratification was carried at 4 °C during three days. Seedlings were grown at a constant temperature of 22 °C under long-day conditions (16 h light 100 µmol m$^{-2}$ s$^{-1}$:8 h darkness).

The Marchantia *ARF* mutants Mp*arf1-4* and Mp*arf3-1* in the Takaragaike-1 accession (Tak-1; male) had been previously described[11,14]. *M. polymorpha* plants were cultured on half-strength Gamborg's B5 pH 5.5–5.8 with 1% agar at 22 °C and constant white light (60 µmol m$^{-2}$ s$^{-1}$), and maintained through asexual reproduction in axenic conditions.

Cultures of *Chlorokybus melkonianii* (previously *atmophyticus*) CCAC 0220, Klebsormidium nitens (previously *flaccidum*) NIES-2285, *Mesotaenium endlicherianum* SAG12.97, and *Penium margaritaceum* Skidmore-8 were grown as indicated by algal collection centres and providers. In general, each strain was grown in species-specific media, either in solid medium (*C. melkonianii, K. nitens*, 2% agar) or liquid (*M. endlicherianum, P. margaritaceum*). Prior to RNA extraction, samples were collected in 15 or 50 ml tubes and freeze-dried.

## Molecular cloning and plasmid construction

Oligonucleotides and plasmids used and generated in this study can be found in Supplementary Data 4 and 5. ARF coding sequences (and specific regions) were amplified from species-specific cDNA, with the exception of *Coleochaete orbicularis, Spirogloea muscicola* and *Spirogyra pratensis* sequences that were amplified from synthesized fragments or plasmids. Gateway™ entry vectors were obtained by transferring PCR-amplified sequences into a linearized pEN207 (amplified with JHG079/080) via NEBuilder® HiFi DNA assemblage (New England Biolabs) unless specified. For most domain swap entries, either pEN207 MpARF1 or MpARF3 entry plasmids were PCR-amplified excluding regions to swap with the corresponding oligonucleotides, and fused via HiFi DNA reaction with suitable PCR-amplified domain sequences. For MpARF3 domain swaps with A- and B-class MpARF domains, a pENTR/D TOPO three-domain swap restriction-ligation strategy was followed as previously[14]. All entry vectors were transferred into the corresponding destination plasmids via LR Clonase II (Invitrogen) reaction.

The plant transcriptional effector destination plasmid pMON-Gal4DBD-GW-mNeonGreen was made by amplifying the Gal4DBD-Gateway cassette fragment from pMpGWB102-GAL4DBD and introducing it into the PCR-amplified backbone of pMON999-mNeonGreen including the 35S promoter and mNeonGreen via NEBuilder® HiFi DNA assemblage (New England Biolabs). The Marchantia endogenous ARF expression destination plasmids pJL001 and pHKDW038 were made as previously described[11,14], except for using pMpGWB301 (Addgene #68629) and pMpGWB307 (Addgene #68635) plasmids as backbone respectively.

*AtARF5* point mutations in the AD/Tudor-like domain were constructed using the pBM8_gARF5 plasmid (*proARF5::ARF5:terARF5*[5],) as scaffold. The region between XhoI and SpeI in pBM8_gARF5, bearing the genomic fragment of *AtARF5*, was replaced with *AtARF5* CDS or the mutated versions (amplified using KT520/KT521 plus the corresponding point mutation combination, including amino acids 3 to 902) through NEBuilder® HiFi DNA reaction.

MpARF1 point mutations in the AD/Tudor-like domain were cloned into a pre-existing gateway entry plasmid. First, the Mp*ARF1* 3'UTR (939 basepairs, amplified using oligonucleotides KT677/678) was introduced into a pEN221-MpARF1 via NEBuilder® HiFi DNA reaction (New England Biolabs) to make the pEN221-MpARF1-3'UTR plasmid. Mutant versions of the CDS were amplified from a wild-type copy using different oligonucleotide pairs and introduced via HiFi DNA reaction into a SpeI/NaeI-cut pEN221-MpARF1-3'UTR fragment.

AD deletions were made by excluding the S326-P392 region for AtARF5 and F286-L369 region for MpARF1. For AtARF5$^{ΔAD}$, the strategy explained above was followed except for using KT826/AFR002 and KT827/AFR001 PCR amplicons using the genomic sequence of *AtARF5* to introduce into pBM8_gARF5. For MpARF1$^{AAD}$, JHG500/502 and JHG503/501 to introduce into pEN207 via HiFi DNA reaction.

Marchantia genome editing plasmids were made as previously described in ref. 38. Briefly, complementary DNA oligonucleotides bearing guide RNA sequences were annealed and introduced into pMpGE_En03 (Addgene #71535), followed by transference into the Cas9-bearing destination plasmid, pMpGE011 (Addgene #71537).

## Arabidopsis transformation and *monopteros* rescue experiments

Heterozygous plants of the *mpB4149* and *mp-12* alleles were transformed by *Agrobacterium tumefaciens*-mediated floral dipping with the corresponding vectors carrying the construct pARF5::ARF5 and its different deleted or mutated versions. T1 generation seeds were selected on half-strength MS with 15 µg/ml phosphinotricin (PPT). T2 generation segregation of the *monopteros* phenotype was checked to determine the mother plant (T1) genotype. The percentage of rootless seedlings observed in the progeny of heterozygous T1 plants was used to determine phenotype rescue degree.

## Marchantia transformation and phenotyping

*Marchantia polymorpha* transformants were obtained by an adapted agrobacterium-mediated thalli transformation protocol based on ref. 39. Briefly, one to four-week old plants were transferred to 0M51C medium, fine-chopped with a sterile razor to small pieces, and transferred to 6-well or 12-well plates. *Agrobacterium tumefaciens* strain GV3101 carrying the appropriate plasmids were inoculated into each well for co-culturing for three days. Washing and plating was performed as the original protocol. Regenerated lines were confirmed by PCR in G1 plants, and, in the case of CRISPR-generated mutations, by sanger sequencing in G1 and G2 plans. To obtain comparable lines expressing different protein versions, Marchantia lines with fluorescently-tagged ARFs were imaged using a Leica SP8X-SMD confocal microscope equipped with a hybrid detector and a pulsed white-light laser (WLL). Citrine was imaged using a water immersion 20X objective (Exc 514 nm, Em 521–569 nm). Z-stacks were obtained for each gemma and two or more lines were chosen based on nuclear fluorescence.

Phenotyping in the Mp*arf1* context was performed growing gemmae in medium supplemented with mock (DMSO) or 3 µM 1-naphthyl acetic acid (NAA) as previously[14]. Thallus area was quantified in thallus halves from independent-growing apical notches after 10 days. To score gametangiophore induction, plants were transferred into inductive conditions of continuous white light (60 µmol m$^{-2}$ s$^{-1}$) supplemented with far-red light (50% power). For Mp*arf3* related phenotypes, gemmae formation was assessed after cup formation by direct visualization under a stereoscope, and growth assays performed by cutting comparable apical regions and quantifying the regenerated area after two weeks.

## RNA isolation, cDNA synthesis, and qRT-PCR analysis

Ten-day old gemmallins were treated with either 3 μM NAA or mock (DMSO) for 1 hour. After this, plants were flash-frozen in liquid nitrogen and grinded into powder using a bead shaker. Total RNA was extracted with a RNeasy Plant Mini Kit (Qiagen) according to the manufacturer's instructions including an on-column DNase I treatment (Qiagen). cDNA was prepared from 1 μg total RNA with a iScript cDNA Synthesis Kit (Bio-Rad) with poly-T oligo following manufacturer instructions. qRT-PCR was performed in a CFX384 Connect Real-Time PCR Detection system (Bio-Rad) with iQ SYBR Green Supermix (Bio-Rad). Oligonucleotide pair HK308/HK309 was used to assess Mp*WIP* (Mp1g09500) expression (Kato et al., 2017). The geometric mean of the reference genes Mp*SAND* (Mp4g07110) and Mp*EF1α* (Mp3g23400)[40] expression was used to normalize expression as previously[41].

## Yeast transactivation assay

Gal4-DNA Binding Domain (BD) fusions were made by transferring the indicated ARF CDSs into the pGBKT7-GW vector from pEN207 entry vectors (see above) via LR Clonase II (Invitrogen). Final GBKT7 plasmids was transformed into the yeast strain Y2HGold and transformants selected on SD medium lacking tryptophan (Trp) using the Frozen-EZ Yeast Transformation II Kit (Zymo Research). Quantitative transactivation tests were performed as previously described (Ref). Briefly, GBKT7-bearing Y2HGold haploid strains were mated with a Y187 haploid strain bearing a disarmed plasmid conferring leucine selection, and selected in SD medium lacking leucine, Trp and uracil. Resulting diploid strains were grown in liquid SD lacking Leu, Trp, and Ura and β-galactosidase activity quantification performed as described in ref. 42 using the substrate ortho-nitrophenyl-β-galactoside (ONPG) for colorimetric measurements.

## Yeast-two hybrid assays

Bait Gal4-DNA Binding Domain fusions were generated as described above with the appropriate PB1-bearing entry vectors. Prey PB1 constructs were generated by fusing PB1 coding sequences to the Gal4-Activation Domain (AD) in pGADT7-GW via LR Clonase II (Invitrogen) from the entry vectors. Strain Y187 was transformed with pGADT7-derived expression vectors, while strain Y2HGold was transformed with pGBKT7 vectors, and selected in SD medium without Leu or Trp, respectively. Diploid strains were produced by mating and selected in SD medium lacking Leu, Trp and Ura. Drop assays were performed in SD medium lacking Leu, Trp and His, in the presence of 3-aminotriazol (3-AT) (Sigma-Aldrich). Quantitative interaction assays were performed in liquid medium by quantifying β-galactosidase activity as previously described in ref. 42.

## Dual luciferase transactivation assays in Arabidopsis protoplasts

Final effector constructs were made by transferring ARF CDSs from entry vectors to the newly made pMON-Gal4DBD-GW-mNeonGreen plasmid (see above) via LR Clonase II (Invitrogen). A previously available pGreenII 5xGal4 UAS dual luciferase construct was used as reporter[42].

Assays were performed by co-expressing the effector plasmids and the reporter plasmid in Arabidopsis (Col-0) leaf mesophyll protoplasts as previously described[43]. Briefly, *Arabidopsis* young leaves were harvested and the epidermal layer was removed using magic tape. Exposed mesophyll cells were released by incubating the leaves with 1% cellulase and 0.2% macerozyme. Protoplasts were isolated and transfected with equal amounts of effector and reporter plasmid via polyethylene glycol-mediated transfection, and incubated for 16 hours. After incubation, protoplasts were collected and lysed using Passive Lysis Buffer (Promega). First, supernatant mNeonGreen-derived fluorescence was measured using a BioTek Synergy H1

Multimode Reader (Agilent) as a protein-level approximated quantification. Next, luciferase activities in the same supernatants were quantified with the Dual-Glo Luciferase Assay System (Promega) using the same reader. Ratio of Firefly luciferase/Renilla luciferase luminescences was relativized to the fluorescence measurements to obtain transactivation activity. Each supernatant was measured three times, and at least four independent transfection events were used per effector to obtain statistical differences between effector activities, as specified in the figure captions.

## Protein expression and purification

PmARF1 and SmARF3 DBD sequences were introduced into PCR-amplified pET His6 MBP TEV LIC plasmid (Addgene #29656) via HiFi DNA reaction (see Supplementary Data 4 for details). Expression of 6xHis-MBP-PmARF1[DBD] and 6xHis-MBP-SmARF3[DBD] fusion proteins was performed as previously described with modifications[5]. MBP-DBD fusion proteins were expressed in *E. coli* strain Rosetta 2(DE3) (Novagen) by inducing with 0.3 mM IPTG for 16 hours at 20 °C. Cell-free extracts were used to purify proteins through two consecutive affinity chromatographies (1st: HisTrap™ High Performance, 2nd: MBPTrap™ High Performance; Cytiva) followed by a size exclusion chromatography (Superdex[R]200 Prep Grade, Cytiva) using an ÄKTA Pure 25 system.[14] MpARF1 and MpARF2-DBD were already available.

## Electromobility shift assays

Assays were done as previously described in ref. 10. Probes were prepared by annealing 5′Cy5-labeled single-stranded DNA oligonucleotides and unlabelled complementary sequences (Supplementary Data 5). Annealing was performed by heating ssDNA oligonucleotide pairs at 95 °C for five minutes followed by slow cool down in a water bath. Protein-DNA mix include 0.09 mg/ml herring sperm competitor ssDNA, 20 nM Cy5-labelled dsDNA, and varying protein concentrations depending on the experiment as detailed in each figure captions, in interaction buffer (20 mM HEPES pH 7.8, 50 mM KCl, 100 mM Tris pH 8.0, 2.5% glycerol, 1 mM DTT). 20 μl of mixes were incubated at 4 °C 1 h and loaded into 2% agarose-0.5X TBE gels. After running, gels were visualized in a Ettan DIGE Imager (GE Healthcare) equipped with a 647 nm excitation and 665 nm emission cube.

## Size exclusion chromatography with multi-angle static light scattering

Purified 6xHis-MBP-PmARF1[DBD] was diluted to 1 mg/ml concentration in 1x phosphate-buffered saline solution (PBS). The sample was then run on a Superdex[R]200 Increase 10/300 GL column (Cytiva) connected to a 1260 Infinity II HPLC system (Agilent) using 1xPBS filtered with a membrane filter (0.1 μm pore size, MF-Millipore™) at room temperature. The molecular mass of the eluted sample was calculated using Multi-Angle Light Scattering with an Optilab 1090 Differential Refractive Index detector (Wyatt Technology) and a miniDawn 1065 Multi-Angle Light Scattering system (Wyatt Technology). The data was analyzed using Astra 8.0 (Wyatt Technology).

## Statistics & reproducibility

Tests, parameters, group sizes and other statistical information are provided in figure captions for each experimental setup. No statistical method was used to predetermine sample size. Sample sizes were determined according to field standards. Experiments were not randomized. Image analysis was done with samples numbered without labelling group genotypes or treatments by different researchers.

## Reporting summary

Further information on research design is available in the Nature Portfolio Reporting Summary linked to this article.

## Data availability

Arabidopsis transgenic lines as well as plasmids generated in the current study are available from the corresponding author upon request. Marchantia lines are not stored, with the exception of the mutant backgrounds, also available upon request. Processed trees can be found in https://itol.embl.de/shared/dolfweijers. Additional data, including raw phylogenetic trees can be found in the Mendeley Data repository dataset. Source data are provided with this paper.

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

## Acknowledgements

We are grateful to team members Hirotaka Kato, Lisa Olijslager, Jan Willem Borst and Sumanth Mutte for help and support, as well as Enrico Scarpella for sharing materials. This work was supported by a Research Grant from the Human Frontiers Research Program (grant RGP0015/2022 to D.W.), by grants from Netherlands Organization for Scientific Research (GSGT.GSGT.2018.013 to J.R., OCENW.KLEIN.027 to D.W. and VICI-865.14.001 to D.W., and OCENW.M20.031 to fund M.d.R.,), by an Overseas Research Fellowship from the Japanese Society for the Promotion of Science to K.T., by grants from the Ministry of science, innovation and universities from the Government of Spain (PID2020-117028GB-I00) and the Agency for Management of University and Research Grants from the Government of Catalonia (2021-SGR-00425) to R.B., and a Marie Skłodowska-Curie Individual Fellowship (MSCA-IF-2020 ref: 101026004 [REOX]) to J.H-G.

## Author contributions

J.H-G. and D.W. conceptualized the study. J.H-G, V.P.C-C. and J.R. did most experiments. K.T. and A.F.R. generated Arabidopsis material. M.D-A. and M.d.R. performed protein expression analyses in Marchantia lines. W.v.d.B. performed protein expression and purification. S.L. performed the SEC-MALS assay. I.C. and R.B. contributed to structural analysis of the ARF-AD and designed AD mutations. J.H-G., V.P.C-C., J.R. and D.W. analysed and discussed data. J.H-G. and D.W. wrote the manuscript with input from all authors.

## Competing interests

The authors declare no competing interests.
