## [Peer Review File · Nature Communications]

REVIEWER COMMENTS

Reviewer #1 (Remarks to the Author):

Evolutionary Origins and Functional Diversification of Auxin Response Factors

Hernández-García et al

The authors propose that (1) the DNA binding domain of ARF transcription factors is derived from an ancient chromatin methyl-histone binding factor combined with a B3 domain; (2) in streptophytes RAV genes share ancestry with ARF genes; (3) PHIP (related to the chromatin domain in ARF proteins) and RAV are not involved in ARF-mediated auxin signalling in land plants; (4) identified an ABC-type ARF in the algal lineages diverging at the earliest node in streptophytes; (5) A ARFs in land plants are neofunctionalized by addition of an activation domain; (6) C ARF function is distinct from that of AB ARF function in land plants and is conserved, at least biochemically, in algae; and (7) propose that the three classes of ARF in land plants (A, B, C) diverged from one another via successive neofunctionalization events. Some of the observations are more novel than others and perhaps these should be highlighted; as there is much data, an attempt to organize views has been divided into the topics below.

(1) the DNA binding domain of ARF transcription factors is derived from an ancient chromatin methyl-histone binding factor combined with a B3 domain:

This is a novel observation; unfortunately, the histone reader aspect is not required in extant ARFs. Do the authors speculate that perhaps the original fusion protein had a reader activity that was subsequently lost and the evolution of the dimerization with two-fold symmetry evolved subsequently? Is it possible that since the AD and DD domains co-evolved with the ARF B3 and the failure of the RAV B3 (see 3 below) is due to its failure to fold/interact properly with the AD/DD?

lines 104-6: do the authors mean the list was 90% ARF proteins? the way it is written suggests that it could mean 90% of known ARF proteins are on the list?

line 144: Kato et al 2020

line 153: delete 'a'

lines 180-1: can the timing of when during streptophyte evolution this occurred? or can this be clarified later as it must have happened in the ancestral streptophyte prior to the formation of the ABC ARF as outlined later?

(2) in streptophytes RAV genes share ancestry with ARF genes

Much of this section is based on previous studies of RAV genes in streptophytes (see below), with the new data being the lack of rescue when substituting the RAV B3 domain for the ARF1 or ARF3 B3 domains and the plausible derivation of AUX/IAA proteins from a RAV ancestor.

lines 184-5, lines 190-2, lines 221-2: this domain architecture of the RAV proteins (and its restriction to charophycean algae and liverworts) was also described in *Marchantia* genome Cell 171, 287–304, Flores et al 2018 *New Phytologist* 218, 1612-1630, and Martin-Arevalillo et al 2019 *PLOS Genetics* 15, e1008400. Flores suggested the RAV as an outgroup for ARF + IAA.

lines 240-241: what does it suggest if MpARF2 and MpARF3 PBI only interacted heterotypically and not homotypically?

(3) PHIP (related to the chromatin domain in ARF proteins) and RAV are not involved in ARF-mediated auxin signalling in land plants

This is novel data — was there any conspicuous phenotype in the PHIP mutants other than a general growth retardation?

(4) identified an ABC-type ARF in the algal lineages diverging at the earliest node in streptophytes

This section clarified many of the remaining questions concerning hypotheses about previously published phylogenies in Mutte et al 2018 *eLIFE* 7, e33399, Flores et al 2018 *New Phytologist* 218, 1612-1630, and Martin-Arevalillo et al 2019 *PLOS Genetics* 15, e1008400.

line 300: clarify 'do neither belong'

line 304: clarify 'clustering behaviour'?

line 307: In Figure 4, the Aux/IAA and nclIAA clades are shown to originate from an ancestral RAV in the ancestral node of streptophytes — does it have to go back this far? why not a later node?

lines 309-10: *Coleochaete* has been reported to have both? in Flores et al 2018 *New Phytologist* 218, 1612-1630

(5) A ARFs in land plants are neofunctionalized by addition of an activation domain

This result is of interest as it might suggest that the ancestral ARFs (A/B and C classes) were both repressors, and that the evolution of the activation in response to auxin arose with the evolution of the AX/IAA-TIR1 perception, and also providing a nice link as to why the two repressive classes (B and C) are essentially independent of auxin.

(6) C ARF function is distinct from that of AB ARF function in land plants and is conserved, at least biochemically, in algae

It is interesting that if C ARF function is so deeply conserved, that the loss-of-function alleles of C ARFs are not that severe, at least in land plants. And while the ABC ARF did not complement, this could be due to a variety of biochemical reasons — leaving open the question of what these algal genes do at a biological level.

(7) propose that the three classes of ARF in land plants (A, B, C) diverged from one another via successive neofunctionalization events

This section provides additional support from the ideas in Kato et al 2020 that the A/B DNA binding domains are largely similar and distinct from that of C ARFs

lines 434-5: grammar — e.g. phenotypes such as flat thalli...?

Finally, in the Introduction and Discussion, a few improvements could be made to clarify a few points and introduce the reader to a broader view of the literature

lines 47-51: should emphasize that it is a subset of ARF factors that participate in auxin responses.

line 55: as this manuscript is focussed on evolutionary questions, one should refrain using phylogenetically ambiguous terms such as plant — what does this mean in this context? land plant? streptophyte? Viridiplantae?

line 66: other references? My brief perusal of the literature suggests this was mentioned earlier in the Marchantia genome paper (Marchantia genome 2017 Cell 171, 287–304)

line 71: perhaps Finet et al (2013 Mol. Biol. Evol. 30, 45–56) should be referenced here, as this was the first paper I could find to define these clades.

lines 75-6: Perhaps the earlier definition of the minimal auxin response should be referenced (e.g. Kato et al 2015 PLOS Genetics 11, e1005084 and Flores et al 2015 PLOS Genetics 11, e1005207)

lines 78-9: C-ARF ref? Mutte et al 2018 eLIFE 7,e33399 and Flores et al 2018 New Phytologist 218, 1612-1630

line 480: Coleochaetophyceae is incorrect; do the authors mean Chlorokybus?

lines 517-521: as this is not a new concept, it is missing references, which should be a smattering of those already listed above dating from 2015.

line 540-1: whose current view, refs? The scenarios proposed here is similar to that in Flores et al 2018 New Phytologist 218, 1612-1630?

Reviewer #2 (Remarks to the Author):

In this manuscript, Hernández-García and colleagues explore in depth how Auxin Response Factors (ARFs) have evolved and diversified in function. To do so they combine phylogenomics, biochemistry, interactions studies (using Y2H approaches) with in vivo analyses (using both whole plants and protoplasts) in the hornwort *M. Polymorpha* and also in the angiosperm *Arabidopsis* to test hypothesis from the phylogenomic analyses using ARFs from a variety of species as well as ARF variants obtained through swaps. This allows the authors to identify a new ancestral class of ARFs, the ABC-ARFs, from which A-, B- and C-ARFs have derived and a plausible ancestral origin of the ARF DNA-binding domain (DBD) from a Tudor-like domain. The amount of work presented in the manuscript is impressive and the data are convincing. The manuscript is also very well written and the authors discuss a very convincing vision of how ARF might have acquired their specific function during evolution. Given the central role of auxin in controlling plant development the results of this manuscript are highly significant not only to plant biologist but also to the many scientists interested in transcriptional regulation.

I have only a few minor concerns that the authors should address:

1- Line 103: why focusing the search on AD domain and not DD ? The authors could justify this.

2- Fig 1d: the B3 domain should be marked on the right similarly to the left part of the panel.

3- Lines 166-169: It would strengthen the demonstration to provide the evidence that the AD deletion should not perturb the B3-DD structure. The authors should also show that this deletion does not trivially lead to degradation of the protein, using western blots for example.

4- Lines 304-306: The specific ARFs mentioned cannot be seen on Fig 2b and f neither are C or AB ARFs. The authors need to modify the figure to make that visible.

5- Line 356: Either I misunderstood the data or "virtually all A-class ARFs ..." is confusing as there is only 3 A-class ARFs that have been used and the 3 of them are activating transcription.

6- Lines 359-361: The authors should say a word about the fact that Sp-ARF weak activation activity is not confirmed in protoplasts.

7- Line 377: If the main trait acquired during the emergence of A-ARFs is transcriptional activation, do the authors suggest that all the other ARFs are repressors ? If it is the case they should test this at least for a few of them.

8 Line 393 - The authors should recall that CmARF is an ABC-ARF. More generally a lot of ARFs from different species are cited throughout the text and it is not easy to remember their identity. The authors could indicate systematically to which class they belong and maybe from which type of organisms they come from. It help the reader.

9- Lines 421-422: "able to bind known AuxREs" should be changed to "able to bind a known AuxRE" as a single one was tested

10- Line 441: Indicate in the text and the figure which are the AB- and ABC-class DBDs (this is in line with comment #8).

11- Page 23: In the figure legend "Figure 4" should be "Figure 8"

Reviewer #3 (Remarks to the Author):

This manuscript describes extensive analyses of divergent relatives of Auxin Response Factors, which mediate transcriptional responses to the plant hormone auxin. Using sequences of algae and other organisms now available, they have reconstructed phylogenies of different domains of these proteins, and set forward new hypotheses as to how the domain structure of ARFs in extant plants arose. In addition they have used domain swap experiments to test functionality and interchangeability of some of the domains from different family members when expressed in the liverwort *Marchantia* (mostly). The work is broad in scope and leads to several interesting conclusions, including that the DNA binding domain includes an ancient repeat of a domain from Tudor chromatin modifying proteins; but that the actual DNA binding arises from insertion of a distinct structure between these motifs; and they have also reconstructed the timing of appearance of different extant ARF classes. I have only a few suggestions for improvement or clarification:

The first panels in Figure 1 could be arranged so as to be more clear. Panels a and b each show the domain architecture, but only one such diagram is needed. The sequence alignment in panel b is so small as to be unreadable, and could perhaps be replaced with some other depiction (since I think the alignment is in the supplemental data anyway).

I gather that the DD and AD portions of ARFs are each similar to the Tudor domain. How similar are AD and DD to each other in sequence and structure?

They conclude that the AD domain is needed for function - while this seems plausible, a more trivial explanation could be that the protein is unstable without that domain. This is a potential problem for any of the "negative" results in the paper.

They postulate (and show in some figure diagrams) a BRD motif in the ARF/RAV ancestor, but I did not understand the rationale for this claim. Do extant ARFs and/or RAVs have such a motif?

In Figure 5a they show several ARFs that fail to complement the *Marchantia arf1* mutant, but the more interesting cases would be those that do complement, pictures of which are only shown in supplemental data. This could be rearranged.

Similarly in figure 5c they show only *Marchantia* and *Arabidopsis* ARFs, but in the text imply that they have conducted a broader survey of ARF gene activation activity. Is that in the supplementary figures?

Response to reviewers

We are grateful for the detailed feedback on our manuscript and for the suggestions for improvement. Essentially all comments relate to matters that need clarification, or better visualization. In individual cases, more information or data was needed. We have considered all comments and made the relevant revisions to our manuscript. We believe that this has improved the clarity of our manuscript and leads to a more balanced presentation. We respond to each point below.

Reviewer #1:

1. *the DNA binding domain of ARF transcription factors is derived from an ancient chromatin methyl-histone binding factor combined with a B3 domain: This is a novel observation; unfortunately, the histone reader aspect is not required in extant ARFs. Do the authors speculate that perhaps the original fusion protein had a reader activity that was subsequently lost and the evolution of the dimerization with two-fold symmetry evolved subsequently? Is it possible that since the AD and DD domains co-evolved with the ARF B3 and the failure of the RAV B3 (see 3 below) is due to its failure to fold/interact properly with the AD/DD?*

We agree with the reviewer that it would have been exciting if the ARF's still held some histone reader activity. However, we believe our data clearly shows that this is not the case. This leaves the (in our opinion) exciting finding that the origin of the ARF DBD domain is in part derived from a histone reader, which is indeed a new finding. We can speculate about the order of events (recombination – dimerization – loss of histone reader activity), but the truth is that we will never know. We therefore decided not to elaborate on this point.

The point about co-evolution between the PHIP-derived AD-DD and RAV-derived B3 in ARFs is excellent, and something we now explicitly mention. Testing this directly would involve a substantial set of biochemical and genetic experiments.

2. *lines 104-6: do the authors mean the list was 90% ARF proteins? the way it is written suggests that it could mean 90% of known ARF proteins are on the list?*

Thanks for noting this. We have now rewritten this sentence to clarify that in the initial PHMMER list, 90% of the hits were ARFs. This can be now found in lines 104 to 108. To help the readers navigate supplementary table 1, we have explicitly commented on the use of the human PHIPs in phmmer (lines 118-119).

3. *line 144: Kato et al 2020; line 153: delete 'a'*

Changed

4. *lines 180-1: can the timing of when during streptophyte evolution this occurred? or can this be clarified later as it must have happened in the ancestral streptophyte prior to the formation of the ABC ARF as outlined later?*

Good point, but we are not sure we can say anything meaningful about this, given the sparse species sampling and sequence availability at the base of the streptophytes. Thus, pinpointing an exact time point for this event is not possible. We therefore refrain from further speculation.

5. *lines 184-5, lines 190-2, lines 221-2: this domain architecture of the RAV proteins (and its restriction to charophycean algae and liverworts) was also described in Marchantia genome Cell 171, 287–304, Flores et al 2018 New Phytologist 218, 1612-1630, and Martin-Arevalillo et al 2019 PLOS Genetics 15, e1008400. Flores suggested the RAV as an outgroup for ARF + IAA.*

We thank the reviewer for pointing this out. All these papers are now properly credited in the section. The only exception is the Marchantia genome paper, where we have not been able to find an explicit comment on RAV architecture. As Martin-Arevalillo et al 2019, Mutte et al. 2018 and Flores-Sandoval et al. 2018 are very specific on this topic, we think these are the proper citations here. As for the Flores et al suggestion, we attempted to recover such an outgroup through phylogenetic reconstruction using the PB1 sequences and, as we reported before in Mutte & Weijers (2020), we could not confirm it. Instead, all our attempts led to RAV+Aux/IAA (in the mentioned manuscript and here). However, this may be due to long branch attraction.

Essentially our observation is, indeed, not new, but we think our thorough trials do strengthen the idea of RAV and ARFs (and Aux/IAA) gene subfamilies being sister lineages.

6. *lines 240-241: what does it suggest if MpARF2 and MpARF3 PBI only interacted heterotypically and not homotypically?*

In the case of MpARF2, this may not make a big contribution, as DBDs also dimerize, which could likely guide or strengthen PB1 homotypic interaction even when PB1 alone might show a lower affinity. In the case of MpARF3, it is difficult to predict, mostly because we do not know if the DBD triggers homodimerization in this class. It is also likely that additional factors may modulate the assembly of higher-order complexes (other proteins, or DNA binding sites, etc). We should add that the biological significance of the B + C class interaction is unclear, as due to their different DBD's, ARF2 and ARF3 will likely not occupy the same chromatin spaces. Even if they did, the outcome would still be recruitment of TPL (i.e.: repression), so it is hard to know if this would have relevant consequences.

7. *...was there any conspicuous phenotype in the PHIP mutants other than a general growth retardation?*

While we do not think this is related anyhow to auxin and/or ARF function, we do show that the gametangiophores of *hip* mutants are heavily affected in terms of development but not in production rate (and this is exactly the opposite of what happens in either *arf1* or *arf3* mutants, which can produce "normal-looking" structures, but have problems in their onset in opposite manners). We also could not see sperm production, for example. We did not test more phenotypes aside of those shown, but all of vegetative development seems to be delayed: production of gemma cups, dichotomous branching. We foresee that analysing these mutants can lead to interesting findings about the unexplored PHIP proteins in eukaryotes, but given the lack of a clear link to ARF function, we consider this outside of the scope of this work.

8. *line 300: clarify 'do neither belong'*

Rephrased. We meant: "neither belongs to A/B, nor to C clades".

9. *line 304: clarify 'clustering behaviour'?*

Rephrased. We meant "behaviour in the clustering"

10. line 307: *In Figure 4, the Aux/IAA and nclIAA clades are shown to originate from an ancestral RAV in the ancestral node of streptophytes — does it have to go back this far? why not a later node?*

This link is merely meant to suggest that Aux/IAA might have arisen from RAVs (which we cannot be certain in any way as of now). However, this type of schematic representation does not include a time scale and should be interpreted without inferring directionality or time points. It should only highlight the presence or absence of certain elements. These are not nodes in a phylogeny, just a roadmap of the family divergence.

11. lines 309-10: *Coleochaete has been reported to have both? in Flores et al 2018 New Phytologist 218, 1612-1630*

It was the only clade showing both indeed, and we do include these sequences in our analyses. However, the majority of species showed this disparity. This is true also for the other species included in the analysis from Flores et al 2018. We have now clarified this in the revision version at lines 310-311.

12. *It is interesting that if C ARF function is so deeply conserved, that the loss-of-function alleles of C ARFs are not that severe, at least in land plants. And while the ABC ARF did not complement, this could be due to a variety of biochemical reasons — leaving open the question of what these algal genes do at a biological level.*

We respectfully disagree. We do consider the phenotype of *arf3* mutants rather severe. It does not produce gemmae and spontaneously produces sexual reproduction structures without the proper environmental cues. This means that the two main forms of reproduction are completely deregulated. In nature, this phenotype would likely be completely selected against.

We agree that the non-complementation can have any number of causes, but given the clear cases of complementation we report with other ARFs, we do believe that this assay allows drawing conclusions regarding functions (non-)equivalence. We made a note to this effect (lines 362-364).

13. lines 434-5: *grammar — e.g. phenotypes such as flat thalli...?*

We do not see the grammatical error, and have left this unchanged.

14. lines 47-51: *should emphasize that it is a subset of ARF factors that participate in auxin responses.*

Done.

15. line 55: *as this manuscript is focussed on evolutionary questions, one should refrain using phylogenetically ambiguous terms such as plant — what does this mean in this context? land plant? streptophyte? Viridiplantae?*

Thank you for catching this. We normally refer to Viridiplantae when using the term “plant”, but we have now corrected this.

16. line 66: *other references? My brief perusal of the literature suggests this was mentioned earlier in the Marchantia genome paper (Marchantia genome 2017 Cell 171, 287–304)*

This has now been included.

17. line 71: perhaps Finet et al (2013 Mol. Biol. Evol. 30, 45–56) should be referenced here, as this was the first paper I could find to define these clades.

We have now cited this paper here.

18. lines 75-6: Perhaps the earlier definition of the minimal auxin response should be referenced (e.g. Kato et al 2015 PLOS Genetics 11, e1005084 and Flores et al 2015 PLOS Genetics 11, e1005207)

Given that we refer to the experimental, functional analysis of the minimal system, we believe that the current reference (Kato et al., 2020) is the correct one. The other papers are cited and acknowledged in this paper.

19. lines 78-9: C-ARF ref? Mutte et al 2018 eLIFE 7,e333399 and Flores et al 2018 New Phytologist 218, 1612-1630

We have now cited these papers here as well.

20. line 480: Coleochaetophyceae is incorrect; do the authors mean Chlorokybus?

Thanks for flagging this, it has now been corrected.

21. lines 517-521: as this is not a new concept, it is missing references, which should be a smattering of those already listed above dating from 2015.

We have corrected this.

22. line 540-1: whose current view, refs? The scenarios proposed here is similar to that in Flores et al 2018 New Phytologist 218, 1612-1630?

We have rephrased this sentence and included the proper acknowledge.

Reviewer #2:

23. Line 103: why focusing the search on AD domain and not DD? The authors could justify this.

This was originally just because of practical reasons: the AD was clearly defined as a continuous sequence within all ARFs, with a pfam/InterPro-related sequence, while the DD mapped to separate regions in the genes, separated by the B3, and in some instances (C-class) by additional disordered and non-conserved sequences. In fact, after all, we now know that it would have led exactly to the same results, as we have later repeated the same search. In fact, we show the use of PHIP cTudor for the search in Supp. Table 1 and the results are again PHIPs and ARFs.

24. Fig 1d: the B3 domain should be marked on the right similiarly to the left part of the panel.

Done.

25. Lines 166-169: It would strengthen the demonstration to provide the evidence that the AD deletion should not perturb the B3-DD structure. The authors should also show that this deletion does not trivially lead to degradation of the protein, using western blots for example.

This is a good point. For exactly this reason, we checked all our lines for expression and nuclear protein localization by means of the fused Citrine protein. This is described in

the M&M. Nuclear localization requires the DBD (but apparently not the AD), and Citrine alone does not specifically target the nucleus (as other FPs). This alone suggests the protein is not being degraded. As for the structure being completely folded, this is almost impossible to test without producing and purifying the protein and doing non-trivial functional tests.

26. *Lines 304-306: The specific ARFs mentioned cannot be seen on Fig 2b and neither are C or AB ARFs. The authors need to modify the figure to make that visible.*

Rather than adding the labels, we have made the trees available through iTOL (<https://itol.embl.de/shared/dolfweijers#>), such that readers can interact with them. We prefer to leave this figure as is to avoid cluttering, just to indicate higher-order phylogenetic relationships.

27. *Line 356: Either I misunderstood the data or "virtually all A-class ARFs ..." is confusing as there is only 3 A-class ARFs that have been used and the 3 of them are activating transcription.*

We have now rephrased this.

28. *Lines 359-361: The authors should say a word about the fact that Sp-ARFc weak activation activity is not confirmed in protoplasts.*

We now comment on this as indicated to the other reviewers. We trust the plant assays more than the yeast assay.

29. *Line 377: If the main trait acquired during the emergence of A-ARFs is transcriptional activation, do the authors suggest that all the other ARFs are repressors ? If it is the case they should test this at least for a few of them.*

This is well established for B class, and TPL interaction has been shown for many of these proteins. It has also been shown for *Chlorokybus melkonianii* ARF (previously *C. atmophyticus*) in Martin-Arevalillo et al. 2018, which we have now placed in the ABC clade. This, by inference, indeed suggests that all these ARFs are repressors.

Testing repression in heterologous systems is non-trivial and most of the time just leads to "non-activation", which is still not a positive result. Proving repression comprises a whole new study on its own, and out of the scope of the current work. We have included a note pinpointing the idea of A class evolving from a repressor state.

30. *Line 393 - The authors should recall that CmARF is an ABC-ARF. More generally a lot of ARFs from different species are cited throughout the text and it is not easy to remember their identity. The authors could indicated systematically to which class they belong and maybe from with type of organisms they come from. It help the reader.*

We tried to do this in the figures using a color code throughout the text, and at first mention, and in the captions. We have not been able to come up with a way to indicate this at every mention without distorting the narrative.

31. *Lines 421-422: "able to bind know AuxREs" should be changed to "able to bind a know AuxRE" as a single one was tested*

Correct, has been modified.

32. Line 441: Indicate in the text and the figure which are the AB- and ABC-class DBDs (this is in line with comment #8).

Corrected.

33. Page 23: In the figure legend "Figure 4" should be "Figure 8"

The whole caption in the embedded Figure 8 was mistakenly swapped for that of Figure 4. The legend for Fig 8 was correct at the end of the text. Thanks for noting this, we have corrected this issue.

Reviewer #3:

34. The first panels in Figure 1 could be arranged so as to be more clear. Panels a and b each show the domain architecture, but only one such diagram is needed. The sequence alignment in panel b is so small as to be unreadable, and could perhaps be replaced with some other depiction (since I think the alignment is in the supplemental data anyway). I gather that the DD and AD portions of ARFs are each similar to the Tudor domain. How similar are AD and DD to each other in sequence and structure?

We thank the reviewer for this suggestion, but we think the rearrangement would hinder specific aspects for more general audiences. The two depictions in 1a and 1b are not showing the same. 1a shows the strict outcome of the PHMMER domain analysis. As such, that depiction is a direct result of the search. However, this is not the complete known and literature-curated domain architecture from ARFs; for example, DD and AD per se are not automatically found in this architecture annotations as they do not exist as Pfam or InterProScan entries. This data is thus introduced in Figure 1b to show the complete view of ARF and PHIP architectures. We think keeping both helps in clarification for the general audience and also for what to be expected from this type of homologous protein searches.

As for 1b, this alignment is again a simplification of a curated alignment. The coloured columns clearly show conservation at both the strict residue identity level and biochemical properties level. In the final (printed) version, the resolution will allow that letters should be visible for people interested, while they allow for zoomed out visualization of just the colours pinpointing the conservation. We rather not use other types of graphical protein conservation visualizations as bar graphs, etc, as they tend to visually exaggerate differences in conserved to non-conserved residues and motifs.

For the second part of the comment, the cryptoTudor domain of PHIP, as was originally defined, has these two subfolds that match the DD and the AD. Both subdomains are independently similar to Royal Family domains: the DD resembles more the Chromodomains, and AD more the Tudor-like domains. Tandem-Royal Family domains are rather common, only this specific Chromodomain+Tudor domain combination is unique to PHIPs (and now, to ARFs) as far as we are aware.

As for the specific question, DD and AD hardly resemble each other in primary structure, but partially show a similar fold. This happens to the plethora of Royal Family members between the different families (as for example, between Chromodomains and Tudor domains). Given that this is outside of the scope and topic of the current work, we do not discuss all these features in detail. Instead, such detailed investigations are part of follow-up work by some of the authors.

35. *They conclude that the AD domain is needed for function - while this seems plausible, a more trivial explanation could be that the protein is unstable without that domain. This is a potential problem for any of the "negative" results in the paper. They postulate (and show in some figure diagrams) a BRD motif in the ARF/RAV ancestor, but I did not understand the rationale for this claim. Do extant ARFs and/or RAVs have such a motif?*

Regarding protein stability: all proteins were expressed as Citrine fusions, and we pre-screened for expression and nuclear localization. This is described in the M&M.

The BRD stands for B3 Repression Domain, as already indicated in lines 188-9. BRDs are well known in both RAVs and ARFs, occasionally referred to as LFG motifs (due to the amino acids that they contain). See Ikeda et al. 2009 and Mutte et al. 2018 for further information. In this context, we acknowledge that in Figure 1, we use the BRD abbreviation for PHIP Bromodomains, we have now just explicitly included the BRD/LFG naming in the text to avoid confusion between Bromodomains and B3 Repression Domains.

36. *In Figure 5a they show several ARFs that fail to complement the *Marchantia arf1* mutant, but the more interesting cases would be those that do complement, pictures of which are only shown in supplemental data. This could be rearranged.*

We have rearranged the figure to showcase these complementing A-class ARFs.

37. *Similarly in figure 5c they show only *Marchantia* and *Arabidopsis* ARFs, but in the text imply that they have conducted a broader survey of ARF gene activation activity. Is that in the supplementary figures?*

No, this refers to work recently published by Morffy et al (2024). We now more explicitly state and cite this.

REVIEWERS' COMMENTS

Reviewer #1 (Remarks to the Author):

My previous comment have been adequately addressed by the authors — the manuscript describes a substantial advance in our understanding of how the ARF transcription factors function in land plants.

Reviewer #2 (Remarks to the Author):

The authors either addressed my points satisfactorily or provide convincing justifications.

However I feel that concerning my former comments #1, #3 & #4 (23, 25 & 26 in their response to the referees), the information should be added in the main text to help the readers.

Reviewer #3 (Remarks to the Author):

In principle they have responded adequately to the previous suggestions. However, I would ask that they re-check that they have actually made all the changes they claim to have made in their response to reviews. In particular, I noticed that two such corrections had not actually been incorporated:

The BRD designation still refers to two distinct domains, the bromodomain (mentioned in line 116 and in Figure 1) and the B3 Repression Domain that is present in RAV and ARF proteins (line 189 and a couple of other places, also in Figure 8). These need to be distinguished by giving them distinct names whenever they occur in the manuscript text or figures. It could also be helpful to define them in the relevant figure legends.

They also mention that they have cited the recent Morffy et al. paper, but there is no discussion of this in the revised manuscript.